# Visual Instruction Bottleneck Tuning

**Changdae Oh     Jiatong Li     Shawn Im     Sharon Li**
Department of Computer Sciences, University of Wisconsin–Madison
{changdae,sharonli}@cs.wisc.edu

## Abstract

Despite widespread adoption, multimodal large language models (MLLMs) suffer performance degradation when encountering unfamiliar queries under distribution shifts. Existing methods to improve MLLM generalization typically require either more instruction data or larger advanced model architectures, both of which incur non-trivial human labor or computational costs. In this work, we take an alternative approach to enhance the generalization and robustness of MLLMs under distribution shifts, from a representation learning perspective. Inspired by *information bottleneck (IB) principle*, we derive a variational lower bound of the IB for MLLMs and devise a practical implementation, ***Visual Instruction Bottleneck Tuning*** (`Vittle`). We then provide a theoretical justification of `Vittle` by revealing its connection to an information-theoretic robustness metric of MLLM. Empirical validation of multiple MLLMs on open-ended and closed-form question answering and object hallucination detection tasks over 45 datasets, including 30 shift scenarios, demonstrates that `Vittle` consistently improves the MLLM's robustness under shifts by pursuing the learning of a minimal sufficient representation. Code: https://github.com/deeplearning-wisc/vittle

## 1   Introduction

In intensive races on the track of frontier-level AI models, we have observed unprecedented achievements through the form of a general-purpose chat assistant known as multimodal large language models (MLLMs) [1, 2, 3, 4] that combine a visual encoder with a large language model. Their universal yet flexible question-answering interface enables MLLMs to easily permeate our lives from general problem-solving [5, 6] to practical applications [7, 8, 9, 10]. While these models may achieve human-like or even surpass human-level performance on certain tasks, a critical gap remains in their robustness—particularly in handling input variations that humans process effortlessly.

Human intelligence thrives on the ability to distill a large amount of sensory and cognitive inputs into concise abstract representations, a process akin to *conceptual compression* [11, 12]. By prioritizing sparse salient features while discarding redundancy, humans can shape a robust prototypical representation of complex data instances that captures a proper level of **invariance to low-level superficial features** for generalization, yet maintains **sensitivity to high-level abstract features** for discrimination [13, 14, 15]. Unfortunately, there are consistent reports implying that the current MLLMs still lag far behind this desired trade-off between invariance and sensitivity [16, 17, 18, 19].

Specifically, MLLMs fail to produce relevant responses under query distribution shifts. That is, they are vulnerable to processing subtly perturbed samples and long-tailed samples [19]. This limitation partially stems from the difficulty of acquiring diverse high-quality multimodal instruction data at scale. When trained using standard maximum likelihood estimation on this relatively limited amount of instruction data, MLLM tends to fit to data-specific patterns and results in a brittle solution [20, 21, 22]. To enhance generalization, existing efforts typically fall into two categories (1) *data-centric* approaches, which collect more instruction data [23, 24, 25] and processes input in a finer granularity [26, 27], and (2) *model-centric* approaches, which scale up the underlying model

39th Conference on Neural Information Processing Systems (NeurIPS 2025).

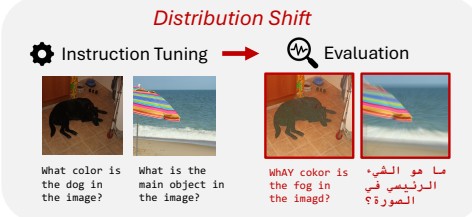
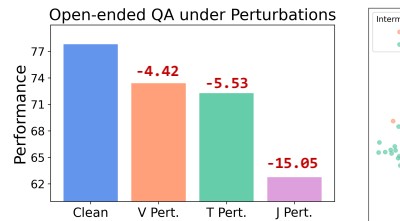
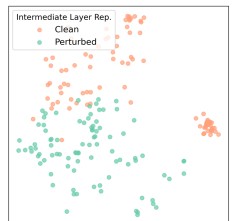

(a) MLLM encounters distribution shifts

(b) Distribution shifts result in MLLM performance drops

Figure 1: **Illustration of distribution shifts for an MLLM (a) and performance degeneration and embedding shifts of the MLLM (b).** An MLLM (LLaVA-v1.5-7B) receives arbitrary queries that might be visually and/or textually perturbed by unexpected noise. These distribution shifts result in performance drops, as shown in the middle bar plot. A visualization of intermediate layer representations of the MLLM on LLaVA-Bench-COCO and its variants indicates that MLLM fails to learn a proper level of invariance to generalize multimodal queries in the representation space.

using more expressive or specialized backbones [28, 29, 30, 31]. However, both data scaling and model scaling are resource-intensive—requiring significant annotation or computational cost.

In this work, we propose a new approach from a *representation-centric* view to improve the robustness of MLLMs under distribution shifts. Rather than scaling data or model, we introduce a lightweight, theoretically grounded module that enhances the internal representations of MLLMs via the information bottleneck (IB) principle. While the IB framework has been explored in small-scale or classification settings [32, 33, 34, 35, 36], integrating it to autoregressive multimodal instruction tuning poses unique challenges due to the complexity of modeling mutual information across high-dimensional, sequential, and heterogeneous modalities. We overcome these barriers by formulating a novel variational lower bound of the IB objective specifically tailored to the multimodal and sequential nature of MLLMs. We further instantiate this formulation as a modular and scalable implementation—***Visual Instruction Bottleneck Tuning*** (`Vittle`), which inserts one simple bottleneck layer within the LLM backbone. `Vittle` pursues *minimal sufficient representations* [37] that try to preserve only response-relevant information while discarding non-essential residual features. To our knowledge, this is the first work to investigate the IB framework for end-to-end instruction tuning of multimodal LLMs, offering a model-agnostic pathway toward building more robust AI systems.

We conduct an extensive evaluation of `Vittle` across a wide spectrum of multimodal benchmarks to assess its robustness and generalization under distribution shift. Our experiments span 30 distribution shifts covering diverse forms of perturbation (in both vision and language) and long-tail distributions. Through these evaluations, we demonstrate that `Vittle` consistently improves robustness over standard instruction tuning baselines, without sacrificing performance on standard benchmarks and canonical tasks. Notably, we find that the bottlenecked representations induced by `Vittle` lead to enhanced invariance in the latent space, aligning semantically similar inputs more closely—even under input shifts—while reducing overfitting to modality-specific artifacts. We also show that `Vittle` is compatible with different MLLMs, offering robustness gains while maintaining similar inference-time cost. These results underscore the practical benefit and theoretical promise of information-regularized representation learning for robust multimodal instruction tuning.

**Contributions**: (1) We propose a new representation-centric framework for improving the robustness of MLLMs under distribution shifts, grounded in the information bottleneck principle. (2) We explore the IB-based end-to-end learning objective of an MLLM for the first time by inducing a new variational lower bound of IB for MLLM and devising a practical instantiation, `Vittle`, supported by theoretical analysis. (3) Through experiments on 30 diverse types of distribution shifts, we thoroughly validate the robustness of MLLMs on open-ended/closed-form QA and object hallucination detection tasks and show advantages of compressive representation induced by pursuing the IB principle.

## 2 Background, Related Work, and Motivation

**Multimodal large language models (MLLMs).** Recent advances in MLLMs integrate a pre-trained language model with a vision encoder through *visual instruction tuning* [38, 39]. To be specific, let $X = (X_v, X_t)$ denote a multimodal input query consisting of visual and textual input, e.g., an image and a corresponding instruction or a question given that image, and $Y$ denote a desired

response given the input query. An MLLM $f_\theta$ with parameter $\theta$ is trained to produce the desired response given an input query with a conditional autoregressive language modeling objective, i.e., $\arg\min_\theta \mathbb{E}_{X,Y}[\sum_{m=1}^{M} \log f_\theta(Y_m|X_v, X_t, Y_{<m})]$ for a sequence of $M$-length responses, where the visual input $X_v$ go through a visual encoder and projector modules to be converted as a sequence of tokens that have the same dimension as text embeddings and can be processed by an LLM backbone[1]. After being trained, these models process a wide array of multimodal instructions to solve arbitrary visual question answering tasks [40].

**Robustness problem in MLLMs.** Despite their impressive performance on standard benchmarks and their growing deployment in real-world applications [8, 41, 42], MLLMs remain vulnerable to input perturbations [43, 44, 45]. For example, MLLMs undergo a systematic performance drop [19] when they encounter samples of superficial perturbations (e.g., varying brightness of image and typo in text) illustrated in Figure 1 (a). As shown in the bar plot of Figure 1 (b), LLaVA-v1.5-7B model undergoes severe performance degradation on LLaVA-Bench-COCO (LB-COCO; [38]) under the perturbations from visual input, textual input, and their joint (V, T, and J Pert.).

We posit that these vulnerabilities arise from the way MLLMs structure their internal representation space. In particular, inputs affected by perturbations are often embedded far from their intact (clean) counterparts, reflecting a distribution shift in the representation space that leads to poor generalization from an information-theoretic perspective [19]. The right side of Figure 1 (b) illustrates this phenomenon: using LLaVA-v1.5, we visualize representations of LB-COCO alongside its challenging variant, where the image and text inputs are perturbed. In this setting, semantically equivalent examples are mapped to distinct and distant regions in the latent space, *suggesting a lack of invariance to superficial input variations, which is crucial for robustness to distribution shifts*.

Motivated by this, our work aims to enhance the robustness of MLLMs by explicitly regularizing their internal representations, encouraging them to retain task-relevant information while discarding input-specific noise—thereby finding a good balance between invariance to low-level superficial features and sensitivity to high-level abstract features for better generalization.

**Information bottleneck principle.** The information bottleneck framework provides a principled approach to measure the quality of representations that are maximally predictive of a target variable while compressing redundant information from an input variable [46, 47]. Numerous works have explored the use of IB training objective [32], across computer vision [48, 49], natural language processing [35, 36], graph learning [34, 50], and time-series modeling [51]. These efforts are supported by theoretical insights suggesting that optimizing for the IB objective can reduce generalization error [33, 52]. However, most prior work focused on classification settings [35, 36] and/or relatively small-scale models [32, 53, 35]. Although a recent study explored IB for MLLMs [54], the authors adopted IB training on a lightweight projector module while keeping the LLM backbone frozen. In contrast, *our work is the first to investigate the IB framework for end-to-end training of large-scale autoregressive multimodal language models*. Beyond shallow adaptations, we directly modify the internal structure of the LLM to promote IB-consistent behavior throughout the training process. We focus specifically on instruction tuning for MLLMs—which have become increasingly central to modern AI ecosystems but remain largely unexplored from the perspective of IB-based learning.

## 3 Method

### 3.1 Preliminary: Information Bottleneck As a Learning Objective

Let $X$ be a multimodal input query (e.g., image-text pair), $Y$ the desired output, and $Z = f(X)$ an intermediate representation extracted by the MLLM encoder $f(\cdot)$. The Information Bottleneck principle aims to learn representations that are *maximally informative about the output $Y$* while being *minimally informative about the input $X$*. Formally, this is expressed as the optimization objective:

$$\max_f \text{IB}_f(X,Y) := \underbrace{I(Z,Y)}_{\text{acquiring information from desired output}} - \beta \underbrace{I(Z,X)}_{\text{compressing input-specific redundant information}}, \quad (1)$$

where $I(\cdot,\cdot)$ denotes mutual information and $\beta$ is the trade-off coefficient. Minimizing $I(Z,X)$ encourages removing redundant or input-specific variations, while maximizing $I(Z,Y)$ ensures that the representation retains task-relevant signals necessary to predict the desired output.

---

[1]For simplicity, we will omit the visual encoder and projector in our learning objective at following sections.

In other words, the IB objective promotes representations that discard non-essential features tied to the input domain, while preserving those critical for solving the task. This property is desirable for robust instruction tuning, where user queries that have the same latent goal must be mapped to consistent responses under varied conditions (e.g., visual and textual perturbations). Despite its appeal, integrating the IB objective into MLLM training is ***highly non-trivial due to the intractability of mutual information estimation and the complexity of autoregressive and multimodal architectures***.

### 3.2 Variational Inference for Information Bottleneck in MLLMs

Directly optimizing the IB objective is generally intractable, as it involves mutual information terms over unknown data distributions. In this work, we introduce a tractable variational bound on the IB objective, specifically tailored to the autoregressive and multimodal structure of MLLMs. We outline the key steps below and provide full derivations in the Appendix D.

We begin with the mutual information term $I(Z, X)$. Given the sequential nature of MLLMs, we decompose both the input $X = (X_v, X_t)$ and the latent representation $Z = (Z_v, Z_t)$ into **v**isual and **t**extual components. We can then derive the following upper bound for $I(Z, X)$:

$$
\begin{aligned}
I(Z, X) = \mathbb{E}_{x,z}[\log \frac{p(z|x)}{p(z)}] &\leq \mathbb{E}_{x,z}[\log \frac{p(z|x)}{r(z)}] = \mathbb{E}_{x_v,x_t,z_v,z_t}[\log \frac{p(z_t|x_v,x_t)p(z_v|x_v)}{r(z_v)r(z_t)}] \\
&= \mathbb{E}_{x_v,x_t}[\mathbb{E}_{z_t|x_v,x_t}[\mathbb{E}_{z_v|x_v}[\log \frac{p(z_v|x_v)}{r(z_v)}]]] + \mathbb{E}_{x_v,x_t}[\mathbb{E}_{z_v|x_v}[\mathbb{E}_{z_t|x_v,x_t}[\log \frac{p(z_t|x_v,x_t)}{r(z_t)}]]] \\
&= \mathbb{E}_{x_v}[D_{\mathrm{KL}}(p(z_v|x_v)||r(z_v))] + \mathbb{E}_{x_v,x_t}[D_{\mathrm{KL}}(p(z_t|x_v,x_t)||r(z_t))], \quad (2)
\end{aligned}
$$

where the first inequality holds given the non-negativity of Kullback-Leibler divergence (KLD), $D_{\mathrm{KL}}(r(z)||p(z))$, and $p(z_v|x_v, x_t) = p(z_v|x_v)$ due to causal mask in MLLM. Here, we introduce $r(z) = r(z_v, z_t) = r(z_v)r(z_t)$ as a factorizable variational approximation of the true prior $p(z)$.

Next, for the output-relevant term $I(Z, Y)$, we have the lower bound as the same as Alemi et al. [32]:

$$
I(Z, Y) = \mathbb{E}_{y,z}\left[\log \frac{p(y|z)}{p(y)}\right] \geq \mathbb{E}_{x,y,z}[\log q(y|z)] - \mathbb{E}_y[\log p(y)] \geq \mathbb{E}_{x,y}\left[\mathbb{E}_{z|x}[\log q(y|z)]\right], \quad (3)
$$

where we replace the true posterior $p(y|z)$ with a variational approximation $q(y|z)$ that will be parameterized by a model component (will be elucidated in Section 3.3).

Finally, combining the lower bound of $I(Z, Y)$ and the upper bound of $I(Z, X)$ yields a variational lower bound for the IB objective as follows,

$$
\begin{aligned}
\mathrm{IB}(X, Y) \geq \ &\mathbb{E}_{x,y}\left[\mathbb{E}_{z|x}[\log q(y|z)]\right] \\
&- \beta\left(\mathbb{E}_{x_v}\left[D_{\mathrm{KL}}(p(z_v|x_v)||r(z_v))\right] + \mathbb{E}_{x_v,x_t}\left[D_{\mathrm{KL}}(p(z_t|x_v,x_t)||r(z_t))\right]\right), \quad (4)
\end{aligned}
$$

In the next section, we elaborate on how we can implement this variational lower bound for MLLM instruction tuning in practice.

### 3.3 `Vittle`: A Practical Implementation of Visual Instruction Bottleneck Tuning

By using a Monte Carlo approximation of expectations over data, Eq. (4) can be written as follows,

$$
\mathcal{L}_\beta = \frac{1}{N} \sum_{i=1}^{N} \mathbb{E}_{z|x^i}[\log q(y^i|z)] - \beta\left(D_{\mathrm{KL}}(p(z_v|x_v^i)||r(z_v)) + D_{\mathrm{KL}}(p(z_t|x_v^i,x_t^i)||r(z_t))\right), \quad (5)
$$

where, $x^i = (x_v^i, x_v^t)$ denotes the $i$-th sample query from an instruction tuning dataset. To compute this empirical estimate of the IB lower bound, we need to model the posterior distributions, $p(z_v|x_v)$ and $p(z_t|x_v, x_t)$, and prior distributions $r(z_v)$ and $r(z_t)$, of the MLLM's inner representation $Z$. Although in principle these distributions can take arbitrary forms, multivariate Gaussian distributions with simplified covariance matrices have been widely adopted in variational inference and probabilistic embedding literature [55, 56, 57, 32, 58, 59, 60, 61] due to their mathematical tractability and empirical effectiveness. By following this common standard, we set the posteriors and priors as Gaussian with diagonal covariance, and will elucidate how exactly they are defined below.

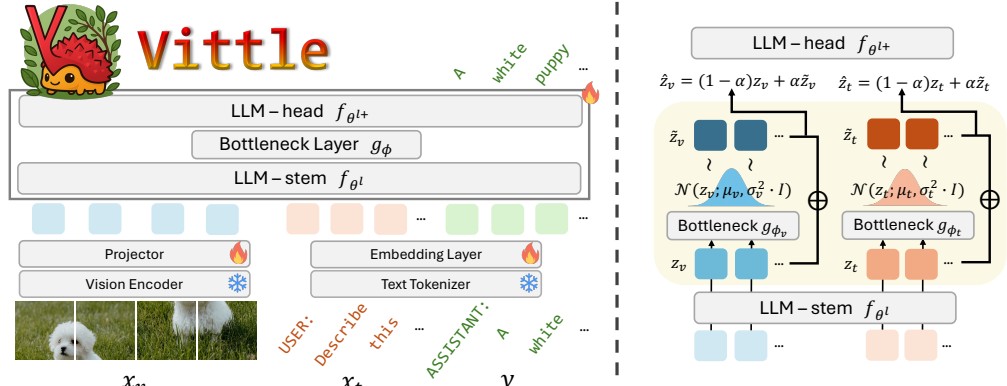

Figure 2: `Vittle` **architecture**. We insert a learnable bottleneck layer $g_\phi = \{g_{\phi_v}, g_{\phi_t}\}$ on top of $l$ blocks of LLM backbone (i.e., LLM-stem $f_{\theta^l}$) to estimate posterior distributions of token embeddings. After obtaining a sample per token $\{\tilde{z}_v, \tilde{z}_t\}$ from posteriors, we interpolate it with a pre-bottlenecked token representation $\{z_v, z_t\}$ and pass it through the remaining LLM blocks (i.e., LLM-head $f_{\theta^{l+}}$).

**Posterior distributions.**   As illustrated in Figure 2, we parameterize the posteriors $p(z_v|x_v)$ and $p(z_t|x_v, x_t)$ using simple feed-forward blocks. Specifically, they are non-linear mappings, $g_{\phi.}$ : $\mathbb{R}^d \to \mathbb{R}^{2d}$ implemented by multi-layer perceptron (MLP) for each modalities, which map each $d$-dimensional token embedding to the posterior Gaussian parameter vectors $\mu \in \mathbb{R}^d$ and $\sigma^2 \in \mathbb{R}^d_+$. Given an intermediate $l$-th layer output representation $(z_v, z_t) = f_{\theta^l}(x_v, x_t)$, we define:

$$p(z_v|x_v) := \mathcal{N}(z_v; \mu_v, \sigma^2_v \cdot I), \quad p(z_t|x_v, x_t) := \mathcal{N}(z_t; \mu_t, \sigma^2_t \cdot I),$$

where $[\mu_v, \sigma^2_v] = g_{\phi_v}(f_{\theta^l}(x_v))$ and $[\mu_t, \sigma^2_t] = g_{\phi_t}(f_{\theta^l}(x_v, x_t))$, with the mean and variance parameters are bipartited along the output dimensions of the MLP. These MLPs are applied position-wise in the same manner as Transformer's feed-forward layers [62], producing token-wise variational posteriors. Now, we can sample from the posterior distributions of MLLM representation by $\tilde{z}_v \sim p(z_v|x_v)$ and $\tilde{z}_t \sim p(z_t|x_v, x_t)$. Then, to strike a balance between invariance and sensitivity, we interpolate the original representation $z$ (pre-bottleneck) with the post-bottleneck counterpart $\tilde{z}$ as $\hat{z} = (1-\alpha)z + \alpha\tilde{z}$[2]. These representations are fed into the remaining layers to compute the predictive distribution over outputs, i.e., $q(y|z) := f_{\theta^{l+}}(y|\hat{z}_v, \hat{z}_t)$. While direct sampling introduces non-differentiability, we can enable the gradient flow using the reparameterization trick [56] to sample $\tilde{z}$ via $\tilde{z} = \mu + \sigma \odot \epsilon$ with $\epsilon \sim \mathcal{N}(\mathbf{0}, I)$ where $\mu$ and $\sigma$ are the outputs of the bottleneck MLP module given input $x$.

**Prior distributions.**   We consider two instantiations of the prior distribution for both $Z_v$ and $Z_t$: (1) a *fixed* standard[3] Gaussian $\mathcal{N}(\mathbf{0}, I)$, which is input-independent and enforces strong isotropy, and (2) a *learnable* Gaussian $\mathcal{N}(\mu_\psi, \sigma^2_\psi \cdot I)$, where $\mu_\psi$ and $\sigma^2_\psi$ are two learnable vectors shared across samples. Each prior affects the formation of representations differently—the fixed prior imposes stronger regularization and robustness, while the learnable prior introduces additional flexibility by allowing the model to adapt to the instruction tuning distribution. We name the former `Vittle` (F) and the latter `Vittle` (L), and validate them altogether for all the evaluations in Section 4.

**Overall objective and implementation.**   The first term of $\mathcal{L}_\beta$(Eq. (5)) can be easily computed through the standard cross-entropy, and our Gaussian instantiation of posteriors and priors allows us to derive closed-form expressions of KLD terms that can be computed from simple arithmetic between $\mu$ and $\sigma^2$ parameters (See Appendix A.2). We set $\beta = \frac{0.1}{d}$ where $d$ is the hidden dimension of the MLLM, to normalize the KL regularization terms relative to the size of the latent dimension. The interpolation coefficient $\alpha$ in $\hat{z} = (1-\alpha)z + \alpha\tilde{z}$ increases progressively following a cosine schedule up to 0.5. During inference, we consistently use an averaged representation $\hat{z} = (z + \tilde{z})/2$ with a deterministic posterior representation $\tilde{z} = \mu$ rather than using a random sample. The target layer to apply the bottleneck module can differ between visual and textual tokens, but we set $l = 24$ for both modalities among 32 layers in a 7B-size LLM, i.e., top 25% layer, by default for simplicity (See Appendix B.1 for the ablation study). Figure 2 depicts the architecture overview.

---

[2]This interpolation yields an asymmetric posterior specification for $I(Z, X)$ and $I(Z, Y)$ (Appendix D).

[3]An alternative is to use the representation statistics (mean and variance) from a pre-instruction-tune model to host informative priors similar to `mixout` [63], which may relax the need for interpolation $(1-\alpha)z + \alpha\tilde{z}$.

## 3.4 Theoretical Justification

The learning objective of `Vittle` has an attractive theoretical interpretation that can support the improvement in robustness of `Vittle`. In this section, we first introduce a recently proposed information-theoretic measure of MLLM's robustness under distribution shifts, ***effective mutual information difference***, EMID [19], and show how `Vittle` can contribute to reducing EMID.

**Definition 3.1** (**EMID**). *Let $P_\Theta : \mathcal{X} \to \mathcal{Y}$ be an MLLM with parameters $\Theta$ that produces an output response $Y_\Theta$ given an input instruction $X$. For joint distributions $P_{XY}$ and $Q_{XY}$, effective mutual information difference of $P_\Theta$ over distributions $P$ and $Q$ is defined as below,*

$$EMID(P_{XY}, Q_{XY}; P_\Theta) := [I(P_{XY_\Theta}) - I(P_{XY})] - [I(Q_{XY_\Theta}) - I(Q_{XY})]. \tag{6}$$

where $I$ denotes the mutual information that measures the relevance between input query and response. A higher value of EMID indicates that MLLM $P_\Theta$ undergoes query-response relavance degeneration in the distribution $Q$ (e.g., test-time) compared to $P$ (e.g., train-time), so we want to achieve a lower value of it to ensure robustness. We now derive an upper bound for the EMID (See Appendix E).

**Proposition 3.2** (**EMID upper bound**). *Let $P_\Theta$ be an MLLM that maps $X = \{X_v, X_t\}$ to $Z = \{Z_v, Z_\Theta\}$, and then subsequently maps $Z$ to $Y_\Theta$. Given joint distributions $P_{XY} = P_X \times P_{Y|X}$ and $Q_{XY} = Q_X \times Q_{Y|X}$ (resp. $P_{ZY}$ and $Q_{ZY}$), by assuming consistent conditionals over $Z_v|Z_t$, $Z_t|Z_v$, and $Y|X$ between $P$ and $Q$, we have an upper bound for $EMID(P_{XY}, Q_{XY}; P_\Theta)$ as below,*

$$\hat{H}\big(D_{JS}^{\frac{1}{2}}(P_{Z_v}||Q_{Z_v}) + D_{JS}^{\frac{1}{2}}(P_{Z_t}||Q_{Z_t}) + \sqrt{\Delta_{X|Z}}\big) + |H(P_{Y_\Theta}) - H(P_Y)| + |H(Q_{Y_\Theta}) - H(Q_Y)|, \tag{7}$$

where $H$ and $D_{JS}^{\frac{1}{2}}$ indicate the entropy and square root of Jensen-Shannon divergence (JSD), respectively, $\Delta_{X|Z} := \mathbb{E}_{z \sim P}[D_{KL}(P_{X|z}||M_{X|z})] + \mathbb{E}_{z \sim Q}[D_{KL}(Q_{X|z}||M_{X|z})]$ with a mixture distribution $M = \frac{P+Q}{2}$, and $\hat{H} := \max_{x \in \mathcal{X}}[H(Q_{Y|x}) + H(P_{Y_\Theta})]$. As we are interested in the terms related to $\Theta$ being optimized, the terms $H(P_Y)$, $H(Q_Y)$, and $\max H(Q_{Y|X})$, can be ignored from Eq. 7 which are fixed across model parameters. We can also ignore $\sqrt{\Delta_{X|Z}}$ term because it cannot directly affect $Y_\Theta$ given the conditional independence $X \perp Y_\Theta | Z$. That is, the upper bound of EMID is boiled down to the multiplication and summation between the output entropy and the representational divergence.

> **Implication.** `Vittle` maximizes the variational lower bound of IB, which consists of (1) minimizing a standard negative log-likelihood term representing an expected risk, and (2) minimizing KLD terms to enforce posterior distributions close to prior distributions. By (1), MLLM $P_\Theta$ seeks a solution $\Theta$ that minimizes the expected risk and reduces its output entropy $H(P_{Y_\Theta})$ and probably $H(Q_{Y_\Theta})$ as well [64, 65, 66]. By (2), it may reduce JSD between representation distributions $P_Z$ and $Q_Z$ by promoting all posterior samples to be laid near the pre-defined priors [67, 68, 69]. In summary, reduced entropy and JSD terms induce a lower EMID, which means that `Vittle` may achieve better robustness to distribution shifts than the standard training method that neglects the divergence in representation space.

We show that `Vittle` indeed reduces empirical JSD and EMID under distribution shifts in Table 4, and demonstrate in Section 4.2 that `Vittle`'s nice theoretical property is translated into consistent robustness gains under 30 distribution shift scenarios while maintaining in-distribution performance.

# 4 Experiment

## 4.1 Setup

**Model and implementation detail.** We adopt LLaVA-v1.5 [70] as our main baseline MLLM, where we set CLIP ViT-L/14-336px [71] as a vision encoder, Vicuna-v1.5-7B [72] as an LLM, and a two-layer MLP as a projector. We follow the standard two-stage training of LLaVA [38], and replicate stage-1 for image-text alignment with the same configuration and dataset (`LLaVA-pretrain-558k`) of LLaVA-v1.5 [70]. Then, on the `LLaVA-mix-665k`, we apply `Vittle` method by inserting the posterior MLP blocks and training the whole model with our IB objective. To validate the scalability and broad applicability, we also consider LLaVA-v1.5-13B, Prism-7B [73], LLaVA-Mini-Vicuna-7B [74], and LLaVA-Llama3-8B-Instruct [75]. Refer to Appendix A for additional details and Appendix B for the results on LLaVA-v1.5-13B and Prism-7B, respectively.

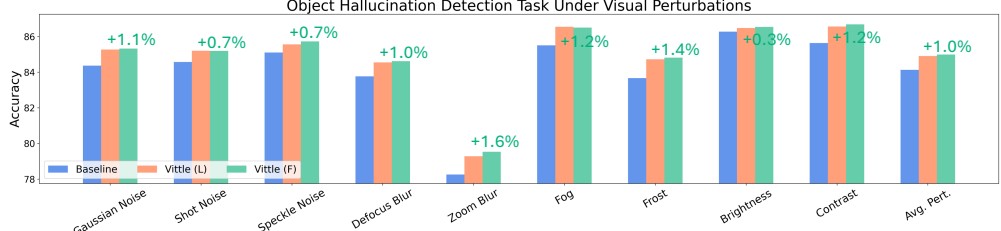

Figure 3: **Object hallucination detection performance on POPE variants**. We enumerate the hallucination detection accuracy of each method on nine versions of perturbed samples, and observe consistent gains by Vittle (highlighted by green numbers of relative improvement from baseline).

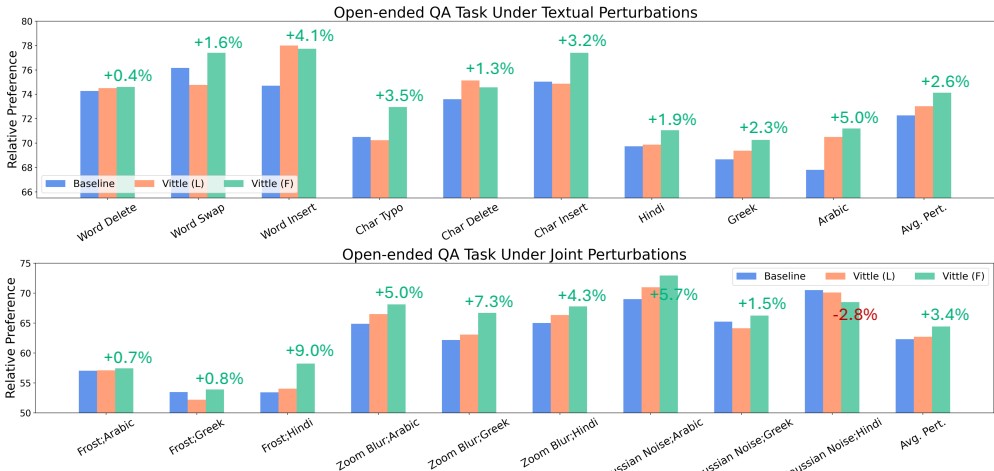

Figure 4: **Open-ended QA performance on LB-COCO variants**. We enumerate the relative preference score of responses from each model on 18 version of perturbed samples, and observe consistent gains by Vittle (especially for the Vittle (F)) on most of the textual (top), and joint (bottom) perturbations (results on visual perturbations are deferred to Appendix B).

**Task, metric, and datasets.** We evaluate instruction-tuned MLLMs with three representative tasks: (1) *open-ended question answering*, (2) *object hallucination detection*, and (3) *closed-form question answering*. All are formatted as a question answering (QA) with a single image input, where we use the average relative preference score measured by GPT-4o LLM judge [76] with three repeated runs for open-ended QA, while using exact matching accuracy for hallucination detection and closed-form QA. For open-ended QA tasks, we adopt four datasets: LB-COCO [38] as a *clean and typical* dataset, and LLaVA-Bench in-the-wild (LB-Wild), LLaVA-Bench-Wilder (LB-Wilder), and WildVision-Bench (WV-Bench) as *long-tail* datasets. Then, we apply 27 types of image and text perturbations on LB-COCO samples[4] to yield 28 variants of *perturbed* LB-COCO (one of clean and nine of visual, textual, and joint perturbations, respectively). For object hallucination detection tasks, we adopt POPE [77] as a *clean and typical* dataset. Then, we generate nine variants of *perturbed* POPE with visual perturbations. Here, we consider the LB-COCO and POPE as in-distribution (ID) datasets because they are generated from MS-COCO samples that construct majorities of the instruction tuning set of modern MLLMs, including LLaVA. For closed-form QA, we adopt four representative datasets: ScienceQA [78], MMMU [79], MME [5], and MMStar [80]. In summary, we experiment with **45 datasets** (31 of open-ended, 10 of object hallucination detection, and 4 of closed-form tasks).

## 4.2 Results

**Vittle improves robustness under input perturbations.** We first evaluate Vittle on object hallucination detection tasks with nine variants of POPE perturbed by visual corruptions in Figure 3. Although MLLMs trained with a standard objective and Vittle similarly suffer from perturbations, two instantiations of Vittle consistently outperform the standard objective. Interestingly, Vittle outperforms the baseline even in clean POPE (See Appendix B).

---

[4]See Appendix A.3 for a comprehensive summary of all perturbations and their generation processes.

We speculate that `Vittle`'s information control prevents the reliance on a partial feature of a single modality [81], e.g., language prior [82], which is a common source of hallucination.

Next, we present the validation on the open-ended QA task with 18 types of input perturbations, which are applied to visual and textual input independently or simultaneously in Figure 4. As we can see, `Vittle` greatly enhances performance in various perturbation datasets highlighted by green numbers that indicate the relative improvements of `Vittle` (F) compared to the baseline. Among the two variants of `Vittle`, `Vittle` (F) showcases better generalization under perturbations than `Vittle` (L), suggesting the benefits of a conservative zero-centered isotropic prior distribution to address a variety of subtle input perturbations. Next, we further explore `Vittle`'s robustness by evaluating varying perturbation severity. To be specific, we generate perturbations on three different degrees that determine how significantly the image or text would be changed.

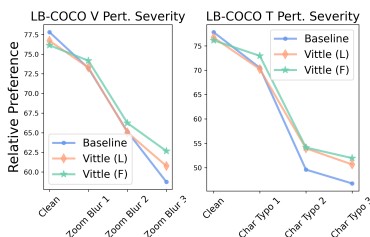

Figure 5: **Evaluation under varying perturbation severity.** `Vittle` achieves better performance, especially on severe perturbations.

In Figure 5, we see that `Vittle` achieves better performance in general, where the margin becomes larger under severe perturbations. In summary, we observe a consistent gain by `Vittle` on the perturbed input setting across two tasks, which indicates that `Vittle` enhances the robustness to distribution shifts by pursuing the minimal sufficiency of data representation.

**Vittle improves generalization to long-tail distributions.** Not only subtle perturbations on input, but long-tail samples are also commonly encountered in many MLLM applications. In Table 1, we validate `Vittle` on three long-tail QA tasks constructed with real-world user queries. We see that `Vittle` also excels in generalizing long-tailed samples compared to the baseline. Interestingly, `Vittle` (L)–learnable prior–exhibits better performance compared with `Vittle` (F). We speculate that a learnable prior IB guides the model to learn a better sensitivity for high-level abstractions as well as an invariance to low-level noise by allowing additional flexibility to shape data-driven priors, yielding superior performance on tasks that require in-depth understanding of irregular queries.

Table 1: **Performance comparison on long-tail open-ended QA tasks** those contain queries that are quite different from typical training samples in terms of visual content and textual semantics.

| Method | LB-Wild | LB-Wilder | WV-Bench |
|---|---|---|---|
| Baseline | 51.6 | 156.9 | 60.0 |
| `Vittle` (L) | **54.6** | **168.8** | **60.4** |
| `Vittle` (F) | 52.2 | 166.1 | 59.7 |

Table 2: **Performance comparison on general benchmark datasets**. These four multi-choice QA datasets require a higher level of multimodal understanding across multiple domains.

| Method | SciQA | MMMU | MME | MMStar | Avg. |
|---|---|---|---|---|---|
| Baseline | 64.6 | **35.6** | 69.7 | **33.7** | 50.9 |
| `Vittle` (L) | 64.7 | 35.3 | **70.5** | **33.7** | **51.1** |
| `Vittle` (F) | **65.4** | 34.5 | 70.1 | 33.5 | 50.9 |

**Vittle preserves competitive performance on general benchmarks.** Although the main focus of `Vittle` is to improve the model's robustness under distribution shifts, securing the rich multimodal understanding capability and knowledge to diverse disciplines is also crucial as an essence of MLLM. To validate this, we evaluate each method on four representative closed-form knowledge-intensive QA benchmark datasets covering various fields. In Table 2, we observe that `Vittle` shows competitive performance with the standard approach, which implies that `Vittle` can also be used as a general-purpose learning objective.

Table 3: **Comparison with weight-space compression methods.** We compare `Vittle` with the LoRA and weight decay (WD) methods on LB-COCO and its perturbed variants.

| Method | Clean | V Pert. | T Pert. | J Pert. |
|---|---|---|---|---|
| Baseline | 77.8 | 73.4 | 72.2 | 62.3 |
| LoRA | 73.4 | 70.4 | 62.7 | 39.7 |
| WD | 74.1 | 72.1 | 73.0 | 59.5 |
| `Vittle` (L) | 76.7 | 73.9 | 73.0 | 62.7 |
| `Vittle` (F) | 76.1 | **74.2** | **74.1** | **64.4** |

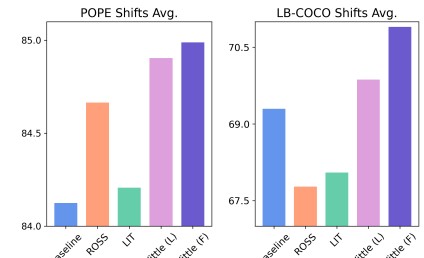

Figure 6: **Comparison with other objective functions.** We report the average performance for all perturbations in POPE and LB-COCO.

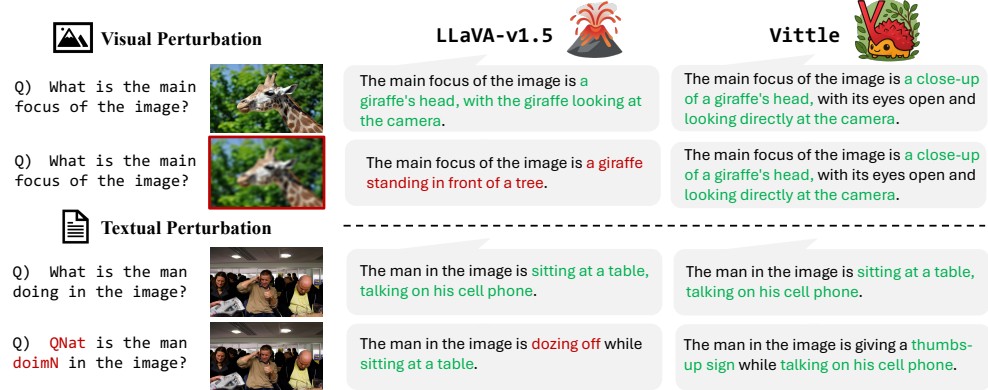

Figure 7: **Case study on LB-COCO under perturbations.** Although LLaVA-v1.5 produces a reasonable response for clean samples, the response and its quality vary under perturbations. Meanwhile, `Vittle` maintains the consistency for the responses.

**Comparison with alternative learning approaches.** Note that the regularization forced by `Vittle` works on the representation space to penalize the amount of information encoded in the data representations. One of the natural alternatives is to regularize the model weight directly. In Table 3, we compare `Vittle` with LoRA [83] and the weight decay method (WD) as instantiations of weight-space regularization, and the results suggest that explicit regularization on weight-space does not ensure a good balance between adaptability on in-distribution and robustness to distribution shifts. The other line of alternatives is *information maximization* during visual instruction tuning [84, 85], which is the exact opposite of `Vittle`'s design principle. We compare two recent methods on this line, ROSS [84] and LIT [85], with `Vittle` on LB-COCO and POPE under perturbations. As shown in Figure 6, although these approaches are effective in improving object hallucination detection performances, they fail to achieve competitive performance on the open-ended QA task (See Appendix B for more results). This implies the non-trivial challenge of devising a general learning objective for MLLMs that can consistently improve robustness across diverse tasks, where we can see the promise of `Vittle` towards broadly applicable robust instruction tuning.

**Qualitative analysis.** Figure 7 shows responses of clean queries and their visually or textually perturbed counterparts. Although the query before and after each perturbation conveys the same meaning and intention, LLaVA-v1.5 reveals volatility in its responses, whereas `Vittle` shows stable behavior by providing consistent responses, i.e., generating exactly the same response in the case of visual perturbation while keep focusing on the same object in the case of textual perturbation.

**Representation analysis.** We next see how `Vittle` shapes the representation space and how it affects robustness. In Table 4, we measure the average value of empirical JSD and EMID discussed in Section 3.4 over 27 perturbed variants of LB-COCO. Both JSD and EMID are computed between two distributions, clean and one of its perturbed versions, and then averaged over 27 clean-perturbed pairs (See Appendix A.4 for details). As our hypothesis, `Vittle` reduces distributional gaps, e.g., achieving smaller JSDs, between clean and perturbed samples in its representation space, thereby achieving a smaller EMID value that indicates better robustness. In Figure 8 (top), we further show PCA visualizations (in the same axis scale) for representations of LLaVA and `Vittle` on clean and image-text perturbed LB-COCO. We see that `Vittle` embeds the clean and semantically equivalent perturbed samples more closely. Moreover, the bottom panel shows that `Vittle` induces smaller cosine distances between clean and perturbed pairs in terms of the histogram and the average value in parentheses. These results indicate that our learning objective is indeed effective in structuring a better representation space that drives robustness.

Table 4: **JSD and EMID evaluation on 27 LB-COCO variants.**

| Method | JSD (↓) | EMID (↓) |
|---|---|---|
| Baseline | 0.068 | 0.026 |
| `Vittle` (L) | 0.048 | 0.021 |
| `Vittle` (F) | 0.047 | 0.025 |

Figure 8: **PCA and pair-wise cosine distance of representations.**

**Cost analysis.** `Vittle` introduces a lightweight bottleneck layer inside of LLM that slightly increases the total number of trainable parameters (by 1.5%). One may thus wonder how `Vittle`'s training and inference time is compared with a bottleneck-free baseline. In Table 5, we show the wall-clock training (per iter and total), and inference time per sample. Although `Vittle` increases the training time up to 20% compared with baseline, *its inference time is almost identical to the original model*, which is a reasonable amount of cost overhead given significant gains in terms of robustness. We also compare the peak memory in Table 14, showing a negligible amount of memory overhead.

Table 5: **Runtime comparison.**

| Method | Tr./it. (s) | Tr. (h) | Te. (s) |
|---|---|---|---|
| Baseline | 7.363 | 11.06 | 0.1048 |
| Vittle (F) | 9.482 | 13.36 | 0.1072 |

**Varying MLLM specification.** As `Vittle` is a model-agnostic learning framework without architecture-specific constraint, we now explore `Vittle` training on two different MLLM specifications in Table 6: (1) LLaVA-Mini that has the same vision encoder and LLM as LLaVA-v1.5 but has different architectural design by incorporating visual token compressor and pre-modality fusion layer; and (2) LLaVA-Llama3-8B-Instruct (denoted in LLaVA++) that we just replace the LLM backbone with Llama3-8B-Instruct. As we can see, `Vittle` consistently outperforms the standard LLaVA training in various MLLM specifications, demonstrating its generality across different model architectures (See Appendix B.5 for more results).

Table 6: `Vittle` **with different model backbones.**

| Backbone | Method | POPE | POPE Shifts Avg. |
|---|---|---|---|
| LLaVA-Mini [74] | Baseline | 79.37 | 77.39 |
| | Vittle (F) | **81.07** | **78.32** |
| LLaVA++ [75] | Baseline | 84.60 | 80.54 |
| | Vittle (F) | **85.87** | **84.08** |

## 5 Conclusion

This work provided the first investigation on the promise of information bottleneck in the context of MLLM instruction tuning to ensure the robustness of MLLM under distribution shifts. We proposed a new theoretically-grounded visual instruction tuning method, `Vittle`. It injects a bottleneck layer inside the LLM to induce posterior samples of internal representations that encode useful information to produce valid responses while discarding other residual information from input queries. With negligible additional cost, `Vittle` is easily optimized with a variational lower bound of IB and shows consistent gains in robustness in 30 types of distribution shifts while also achieving competitive performance on standard benchmarks, indicating that `Vittle` promotes a good balance between invariance and sensitivity during representation learning.

**Limitation and future work.** One possible concern with `Vittle` is its reliance on the quality of $Y$, i.e., a gold response to given instruction, which is usually generated by another LLM. As disclosed by Yeh et al. [86], existing datasets for supervised fine-tuning are quite noisy, and we cannot ensure the advantage of IB on this noisy annotation setup. Moreover, IB alone does not guarantee the generalization of multi-modal counterfactual samples with conflicting language prior [87, 82] or samples from completely different domains [88, 89, 90], and may require additional annotations. Besides, we observed that `Vittle` slightly hurts the optical character recognition capability of MLLM, indicating the importance of further exploration to preserve the fine-grained recognition capability while pursuing robustness to distribution shifts. Investigating the potential of noisy annotation, counterfactual/domain generalization, and fine-grained recognition setups can be interesting future research problems. Meanwhile, the noise-robust representation space achieved by IB training can be helpful for representation steering methods [91, 92] that are also worth exploring for future work.

## Acknowledgments and Disclosure of Funding

The authors thank Seongheon Park and Sean Xuefeng Du for their valuable suggestions and discussions that shaped the draft, and thank all the NeurIPS 2025 PCs, SACs, ACs, and anonymous reviewers. In addition, Changdae Oh particularly thanks Sanghyuk Chun for his sharp feedback on the manuscript. Research is supported in part by the AFOSR Young Investigator Program under award number FA9550-23-1-0184, National Science Foundation (NSF) Award No. IIS-2237037 and IIS-2331669, Office of Naval Research under grant number N00014-23-1-2643, Open Philanthropy, Schmidt Sciences Foundation, and Alfred P. Sloan Fellowship.

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

# Appendix

## Contents

## A   Extended Experiment Setup and Implementation Detail

### A.1   Model and Training

In this work, we consider LLaVA-v1.5 [70] as our target multimodal large language model (MLLM) with CLIP ViT-L/14-336px [71] and Vicuna-v1.5-7B [72] as visual encoder and LLM backbone, respectively, and a two-layer MLP as projector (modality connector that maps features of the visual encoder into the text embedding space). Although all of the results presented in the main body of the paper were produced with LLaVA-v1.5-7B, we also experimented with LLaVA-v1.5-13B (with Vicua-v1.5-13B as the LLM backbone) to validate the scalability of our method, and consider Prism-7B [73] as an additional MLLM architecture to validate the broad applicability of `Vittle`. For fair comparison, all models are trained on the `LLaVA-pretrain-558k` and `LLaVA-mix-665k` datasets, consisting of a mixture of LAION [93], CC [94], SBU [95] datasets with BLIP captions [96] and a mixture of `LLaVA-instruct-158K` and academic-task-oriented (V)QA datasets, respectively. Training configurations such as optimizer, learning rate, and batch size are summarized in Table 7 and Table 8. All training runs are conducted with eight A100-80GB GPUs with DeepSpeed ZeRO library. The shortest run takes roughly 11 hours, whereas the longest run takes about 14 hours. Now, we elaborate on the overall workflow of LLaVA and Prism below.

Table 7: **Hyperparameter list of `Vittle` training.** We adopt exactly the same configurations with LLaVA-v1.5 [70] for Stage 1 and 2.

| Config | Stage1 | Stage2 |
|---|---|---|
| Global batch size | 256 | 128 |
| Batch size per GPU | 32 | 16 |
| Learning rate | 1e-3 | 2e-5 |
| Learning rate schedule | Cosine decay w/ linear warmup | |
| Warmup ratio | 0.03 | |
| Weight decay | 0.0 | |
| Epoch | 1 | |
| Optimizer | AdamW | |
| Precision | bf16 | |

Table 8: **Hyperparameter list of `Prism-Vittle` training.** We adopt exactly the same configurations with Prism-DINOSigLIP-Controlled-7B [73] single stage training.

| Config | Value |
|---|---|
| Global batch size | 128 |
| Batch size per GPU | 16 |
| Learning rate | 2e-5 |
| Learning rate schedule | cosine decay w/ linear-warmup |
| Warmup ratio | 0.03 |
| Weight decay | 0.1 |
| Epoch | 1 |
| Optimizer | AdamW |
| Precision | bf16 |

**LLaVA** is built with a pre-trained visual encoder that takes visual inputs, a pre-trained LLM backbone that takes text instructions, and a lightweight projector that maps features produced by the visual encoder into the text embedding space of LLM backbone so that the visually-grounded multimodal instruction input query can be processed by the LLM backbone. LLaVA undergoes a two-stage training: (1) The first stage takes into account modality alignment, where the projector is trained on image and corresponding instruction or caption with a conditional language modeling loss implemented by aggregating cross-entropy losses across response tokens, while the visual encoder and LLM backbone are frozen. (2) The second stage stands for the instruction tuning, where the projector and LLM backbone are jointly trained on multimodal instruction samples with the same conditional language modeling loss while the visual encoder is still frozen. This two-stage training has been considered a standard approach for developing MLLMs and is widely adopted [97, 98, 29].

**Prism** has a model architecture similar to LLaVA, but provides some valuable insight into the design of the MLLM training recipe, and we note two remarkable design choices of Prism that distinguish it from LLaVA: (1) incorporating multiple visual encoders rather than hosting a single visual encoder, and (2) reducing the two-stage alignment-then-instruction tuning into a single-stage instruction tuning. Note that different self-supervised visual representation learning induces features that have different strengths, and several works reveal the benefits of ensembling multiple different visual encoders to leverage complementary advantages [99, 73, 30]. Prism incorporates SigLIP [100] and DINOv2 [101] to enjoy both a robust global feature and a fine-grained local feature. Meanwhile, Karamcheti et al. [73] showed that the simplified single-stage training strategy can be a cost-effective alternative to the standard two-stage training.

To train these MLLMs, we consider five baseline approaches: (1) the standard full LLM fine-tuning with conditional language modeling loss, (2) parameter-efficient LoRA [83] fine-tuning with the conditional language modeling loss, (3) conditional language modeling loss with weight decay regularization, (4) reconstructive visual instruction tuning (ROSS) [84], and (5) learning to instruct (LIT) [85]. For the LoRA-based training configuration, we use the same one provided by the official LLaVA-v1.5 repository[5], and for the weight decay regularization, we select the regularization magnitude parameter among $\{0.1, 0.01, 0.001\}$ based on the POPE evaluation result. We now elucidate two competitive baseline methods, ROSS and LIT, in the following paragraphs. It is worth noting that these methods are designed to encode more (visual) information into the representation space, which is opposite to our `Vittle`'s design motivation that pursues a minimal sufficient representation for improving robustness to distribution shifts.

**Reconstructive visual instruction tuning (ROSS)** follows the two-stage training of LLaVA, but tries to reconstruct the visual inputs from the LLM backbone by adopting a regression or denoising learning objective in addition to language modeling loss during its second stage. By doing so, ROSS guides the MLLM to learn a much richer visual understanding, which is usually lacking in modern MLLMs [99, 81]. The reconstruction target can be a raw RGB pixel value or the latent representation from an external visual encoder such as VQGAN [102] or VAE [56], and ROSS requires an additional trainable module to reconstruct visual content, which is discarded during inference. We follow the training recipe from the official code repository[6] to replicate ROSS-D-7B with the same visual encoder and LLM backbone to LLaVA-v1.5. For a fair comparison with LLaVA and `Vittle`, we

---

[5]https://github.com/haotian-liu/LLaVA
[6]https://github.com/Haochen-Wang409/ross

train the ROSS with the same dataset (that of LLaVA-v1.5) for both training stages, while the original ROSS model was trained on a slightly larger dataset in the second stage.

**Learning to Instruct (LIT)** also focuses on the visual shortcomings of current MLLM and tries to improve the visual understanding capability of MLLM by incorporating an additional loss term that incentivizes the encoding of additional visual information. To be specific, while the cross-entropy loss in LLaVA's conditional language modeling objective is aggregated through the response tokens only, LIT introduces an extra cross-entropy loss term, which is aggregated over the instruction (question) tokens only, thereby enforcing MLLM to learn to predict a proper textual instruction given an image. As LIT uses the same visual and language backbone model and training dataset as LLaVA-v1.5, we use the pre-trained checkpoint of LIT from Hugging Face[7] for evaluation.

In Figure 6, we observe that while ROSS and LIT are somewhat effective in improving performance on object hallucination detection tasks with the aid of enhanced visual understanding capability, they significantly underperform `Vittle` and even the original LLaVA on the open-ended QA task under distribution shifts. This implies that pursuing more information encoding during visual instruction tuning may not result in better robustness to distribution shifts, but aiming to learn a minimal sufficient representation via `Vittle` can be a promising solution for this (See Table 12 for details).

## A.2 `Vittle` Implementation Details

This section provides additional details on implementing `Vittle` through Python-style pseudo code in Figure 9 and text below. Following the standard two-stage LLaVA training recipe, we freeze the visual encoder and LLM backbone during the first stage and only train the projector module. In the second stage, `Vittle` inserts a bottleneck layer $g_\phi$, consisting of two of the two-layer MLPs $\{g_{\phi_v}, g_{\phi_t}\}$ for visual and textual modalities, inside the LLM backbone to estimate the distributional parameters (mean and diagonal covariance) of the posterior Gaussian distributions for each visual and textual token. Each bottleneck module is constructed with $\{$nn.Linear(d,d), nn.GELU(), nn.Linear(d,2*d)$\}$ where d denotes the hidden dimension of the LLM backbone, and this results in a slightly increased number of model parameters (up to 1.5% from the baseline). We use these estimated distribution parameters to sample a representation from this posterior via $\tilde{z} = \mu + \sigma \odot \epsilon$ where $\epsilon \sim \mathcal{N}(\mathbf{0}, I)$. Then, for a given bottleneck layer index $l$ and for the maximum length of visual $M_v$ and textual input tokens $M_t$, the bottleneck layer $g_\phi$ takes a sequence of token representations $z = \{z_{v,1}, ..., z_{v,M_v}, z_{t,1}, ..., z_{t,M_t}\}$ produced from the layer $l$ to build information-penalized representations $\hat{z} = \{\hat{z}_{v,1}, ..., \hat{z}_{v,M_v}, \tilde{z}_{t,1}, ..., \tilde{z}_{t,M_t}\}$, where $\hat{z} = (1 - \alpha)z + \alpha g_\phi(z)$. Here, we use an interpolated representation between the original pre-bottleneck representation $z$ and the post-bottleneck representation $g_\phi(z)$ with an interpolation coefficient $\alpha$ that progressively grows from 0 to 0.5 by a cosine schedule during training. We observe that solely using the post-bottleneck representation induces a diverging language modeling loss at the later steps of training, and speculate that it is hard to generate a valid response with the information-penalized representation only[8].

Then, we jointly train the LLM backbone, the projector, and this bottleneck layer together during the second stage of training with the objective function 5. As we assume a diagonal covariance Gaussian for the prior and posterior distributions, Kullback–Leibler divergence (KLD) between the prior $p$ and posterior $q$ can be easily expressed as below,

$$D_{\mathrm{KL}}(q, p) = \frac{1}{2} \sum_{j=1}^{d} \big( \log \frac{\sigma_p^2[j]}{\sigma_q^2[j]} - 1 + \frac{(\mu_p[j] - \mu_q[j])^2}{\sigma_p^2[j]} + \frac{\sigma_q^2[j]}{\sigma_p^2[j]} \big) \tag{8}$$

where $\mu.$ and $\sigma.$ denote $d$-dimensional distributional parameter vectors and $[j]$ indicates $j$-th element from the vectors. `Vittle` has two important hyperparameters: (1) target layer index $l$ for bottleneck application, and (2) posterior KLD regularization strength parameter $\beta$. After tuning across $l \in \{24, 28, 31\}$ and $\beta \in \{\frac{0.01}{d}, \frac{0.05}{d}, \frac{0.1}{d}, \frac{0.2}{d}, \frac{1.0}{d}\}$ where $d$ denotes the latent dimension of the LLM backbone, we set $l = 24$ and $\beta = 0.1/d$ based on the average performance of POPE and LB-COCO clean datasets. The interpolation coefficient $\alpha$ can also be tuned, but we found that increasing $\alpha$ beyond 0.5 hinders stable training and observing increased language modeling loss at the later parts of training progress. Figure 12 and Table 11 present the results of hyperparameter ablation study.

---

[7] https://huggingface.co/zhihanzhou/LIT-LLaVA-1.5-Vicuna-7B/tree/main

[8] We suspect that this is because the initial parameter of MLLM's instruction tuning are far from random standard Gaussian or zero therefore induces unstable learning signal from the KLD term.

```
def reparam(mu, logvar):
    std = (logvar / 2).exp()
    batch_size, seq_len, hidden_dim = mu.shape
    z = torch.randn(batch_size, seq_len, hidden_dim)
    return mu + std * z

def forward(self, input_embeds, img_seq_len, a, **kwargs):
    ...
    hidden_states = input_embeds
    for l_idx, llm_layer in enumerate(self.llm.layers):
        layer_outputs = llm_layer(hidden_states, **kwargs)
        hidden_states = layer_outputs[0]
        if l_idx == self.bottleneck_layer_idx:
            # posterior inference
            v_params = self.g_v(hidden_states[:,:i_seq_len,:])
            t_params = self.g_t(hidden_states[:,i_seq_len:,:])
            v_mean = v_params[:,:,:self.h_dim]
            v_logvar = v_params[:,:,self.h_dim:]
            t_mean = t_params[:,:,:self.h_dim]
            t_logvar = t_params[:,:,self.h_dim:]
            v_post = reparam(v_mean, v_logvar)
            t_post = reparam(t_mean, t_logvar)
            z_post = torch.cat((v_post, t_post))
            # interpolation between original and bottlenecked
            hidden_states = (1-a) * hidden_states + a * z_post
    ...
```

Listing 1: Forward pass of `Vittle`

```
def normalized_kld(mu, logvar, modality=None):
    if modality is None:
        # vittle (F) - fixed prior N(0,I)
        kl_loss = -0.5 * (1+logvar-mu**2-logvar.exp()).mean()
    else:
        # vittle (L) - learnable prior
        mu_pr, logvar_pr = self.l_prior[modality]
        logvar_d = logvar-logvar_pr
        scaled_mu_d = (mu-mu_pr).pow(2)/logvar_pr.exp()
        var_ratio = logvar.exp()/logvar_pr.exp()
        kl_loss = -0.5 * (1+logvar_d-scaled_mu_d-var_ratio).mean()
    return kl_loss

def loss(self, logits, labels, v_mean, v_logvar, t_mean, t_logvar):
    lm_loss = self.llm.loss_function(logits, labels)
    if self.learnable_prior:
        flag_v, flag_t = "v", "t"
    else:
        flag_v, flag_t = None, None
    kld_v = self.normalized_kld(v_mean, v_logvar, flag_v)
    kld_t = self.normalized_kld(t_mean, t_logvar, flag_t)
    return lm_loss + self.beta * (kld_v + kld_t)
```

Listing 2: Training objective of `Vittle`

Figure 9: PyTorch-style pseudo code for the forward pass and training objective of `Vittle`

## A.3 Downstream Task Benchmark Construction

**Open-ended QA task.** One of the most representative applications of an MLLM is the generation of free-form responses given multimodal instruction queries. We consider LLaVA-Bench COCO (LB-COCO; [38]) as a typical in-distribution (ID) open-ended QA dataset, which is constructed from MS-COCO images [103] with GPT-generated text queries that have 90 pairs of image and text. We then generate 27 variants of this LB-COCO by applying nine types of image perturbations, nine types of text perturbations, and nine types of image-text joint perturbations, to benchmark MLLMs' robustness under various distribution shifts (which will be elaborated at the end of this section). Meanwhile, we also consider three datasets LLaVA-Bench in-the-wild (LB-Wild; [38]), LLaVA-Bench Wilder (LB-Wilder; [104]), and WildVision-Bench (WV-Bench; [105]) constructed by real-world web users' image-text paired queries of 60, 128, and 500 samples, respectively, to validate models' capability to address long-tailed queries in practice. This results in 31 different open-ended QA datasets in total: clean and 27 perturbed LB-COCO variants, and three long-tail datasets.

**Object hallucination detection task.** Meanwhile, one of the most crucial evaluation aspects of an MLLM is the degree of hallucination of its output. A representative benchmark for this is the POPE dataset [77], where the model is tasked to answer in binary {Yes, No} form given a question about the object's existence given an image. The POPE dataset was also created from the MS-COCO source images with 9,000 corresponding questions, and we consider this dataset as an ID dataset. As we did for the LB-COCO dataset, we generated 9 variants of POPE by applying nine types of visual perturbations to images. Textual perturbations were not considered here because the text query of this dataset is relatively short, so perturbing the core object word token can distort the desired semantics of the question. In summary, we conducted validation on 10 different POPE variants.

**Closed-form QA task.** There are numerous closed-form QA datasets that assess the internal knowledge of MLLMs from various perspectives. In this paper, we consider four representative datasets: ScienceQA [78], MMMU [79], MME [5], and MMStar [80], which are designed to validate multimodal knowledge and understanding capability across various domains.

**Distribution shift simulation.** The goal of this study is to improve the robustness of MLLM under distribution shifts. We mainly focus on subtle perturbations on image and text, which is worth-noting problem given the fact that current MLLMs undergo systematic performance degradation under perturbations. We consider nine visual perturbations listed in Table 9, nine textual perturbations listed in Table 10, and nine image-text joint perturbations: {zoom_blur, frost, gaussian_noise} × {arabic, greek, hindi}. The translations for Arabic, Greek, and Hindi languages from English are conducted by OpenAI GPT-4o with a prompt: "Please translate a {SOURCE} sentence provided by the user into {TARGET}.", and all the remaining perturbations are generated MMRobustness source code[9]. The actual examples of each visual textual perturbation are presented in Figure 11 and Figure 10.

Table 9: **List of visual perturbations.** We consider nine visual perturbations from four categories: (1) Blur, (2) Digital, (3) Weather, and (4) Noise, to validate the robustness of MLLMs under diverse types of visual perturbations.

| Name | Category |
|---|---|
| Defocus Blur | Blur |
| Zoom Blur | Blur |
| Contrast | Digital |
| Brightness | Weather |
| Fog | Weather |
| Frost | Weather |
| Gaussian Noise | Noise |
| Shot Noise | Noise |
| Speckle Noise | Noise |

Table 10: **List of textual perturbations.** We consider nine textual perturbations from three categories: (1) character-level, (2) word-level, and (3) sentence-level, to validate the robustness of MLLMs under diverse types of textual perturbations.

| Name | Category |
|---|---|
| Char Typo | Character-level Perturbation |
| Char Delete | Character-level Perturbation |
| Char Insert | Character-level Perturbation |
| Word Swap | Word-level Perturbation |
| Word Delete | Word-level Perturbation |
| Word Insert | Word-level Perturbation |
| Arabic Translation | Sentence-level Perturbation |
| Greek Translation | Sentence-level Perturbation |
| Hindi Translation | Sentence-level Perturbation |

[9] https://github.com/Jielin-Qiu/MM_Robustness

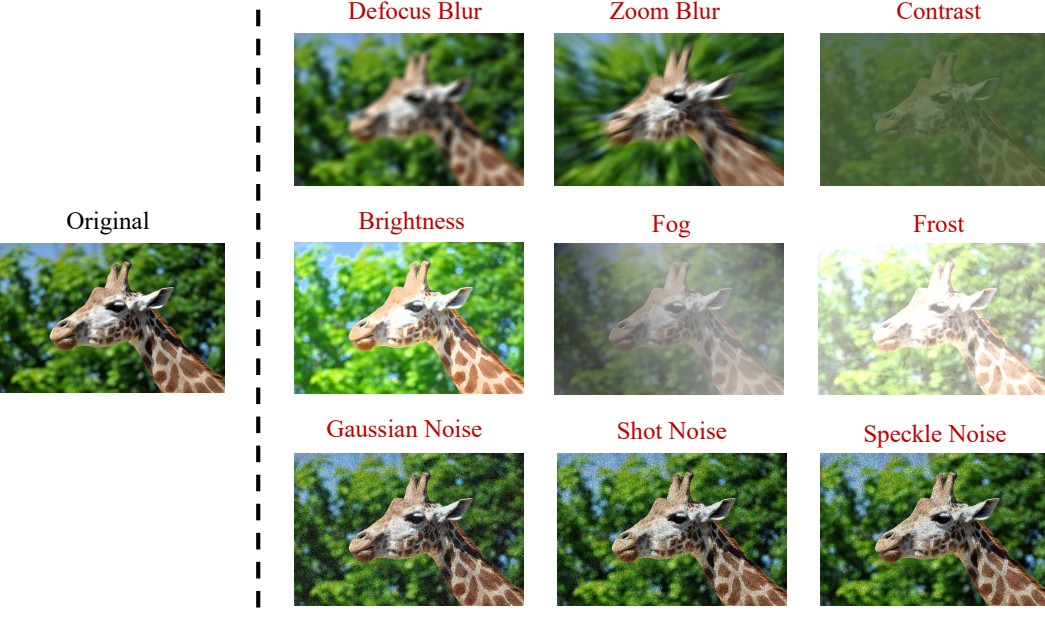

Figure 10: **Examples of visual perturbations.**

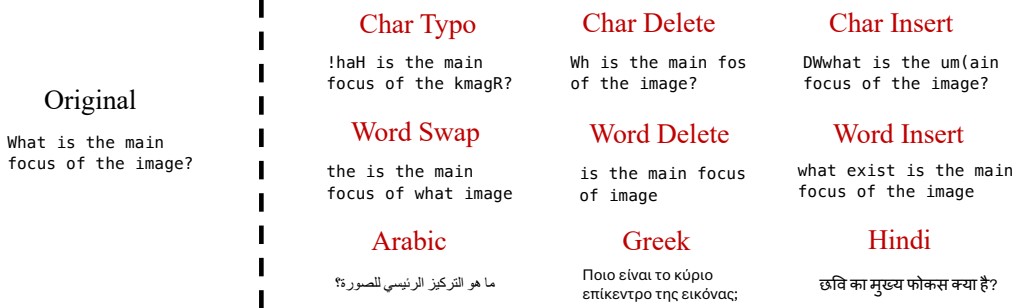

Figure 11: **Examples of textual perturbations.**

## A.4 Evaluation Details

**Open-ended QA task.** Compared to multi-choice closed-form QA tasks that have a unique ground-truth answer per question, open-ended free-form generation-style QA tasks do not provide a single ground-truth answer. We follow the current standard evaluation paradigm, (M)LLM-as-a-Judge, that uses an external (usually more powerful) MLLM to gauge the quality of our target MLLM of interest via prompting. To be specific, for a given input query $x$, reference answer $y$, MLLM $f_\theta : \mathcal{X} \to \mathcal{Y}$, and the judge model $r : \mathcal{X} \times \mathcal{Y} \to \mathbb{Z}^+$, *relative preference* score is defined as, $\mathbb{E}_{x,y}\left[\frac{r(x,f_\theta(x))}{r(x,y)}\right]$.

For all of our open-ended QA evaluations, we used the same system prompt template provided by LLaVA authors[10], and we also adopted the MS-COCO annotation[11]-based GPT-4 response[12] and the `gpt_answer`[13] released by LLaVA-NeXT authors as reference answers for LB-COCO variants and LB-Wilder, respectively. For LB-Wild and WV-Bench, we generated reference answers with GPT-4o.

---

[10]https://github.com/haotian-liu/LLaVA/blob/main/llava/eval/table/rule.json
[11]https://github.com/haotian-liu/LLaVA/blob/main/llava/eval/table/caps_boxes_coco2014_val_80.jsonl
[12]https://github.com/haotian-liu/LLaVA/blob/main/playground/data/coco2014_val_qa_eval/qa90_gpt4_answer.jsonl
[13]https://huggingface.co/datasets/lmms-lab/LLaVA-Bench-Wilder

**Object hallucination detection and closed-form QA task.** In contrast to open-ended tasks, all object hallucination detection and closed-form QA tasks provide a single ground truth answer as a form of discrete labels such as {Yes, No} and {A, B, C, D, ...}. For the multi-choice QA datasets, MMMU, MMStar, and ScienceQA, we attached a subfix prompt: "Answer in a character from the given choices directly." at the end of each question for answer formatting, while using the original question text for YES-or-NO datasets, MME and POPE, without a formatting prompt. We measured the exact matching accuracy $\mathbb{E}_{x,y}[\mathbb{I}(\theta(x) = y)]$ for these tasks.

**Effective Mutual Information Difference (EMID) and Jensen-Shannon Divergence (JSD).** In addition to the evaluation with traditional metrics, we also consider the EMID and JSD-based evaluation, which was recently proposed as an information-theoretic approach to measure the robustness of MLLMs [19]. To compute the empirical estimates of MI, which is required for EMID computation, we use the CLUB estimator [106] and reproduce the training and inference process of [19] by adopting image and text embeddings for the input image and text from CLIP-ViT-B/32 [71] and XLM-RoBERTa-Base [107] to replace $X_v$ and $X_t$, and also the text embeddings of XLM-RoBERTa-Base for responses $Y$ and $Y_\Theta$. To compute empirical estimates of JSD, we adopted the representation JSD estimator [108] on top of the CLIP-ViT-B/32 and XLM-RoBERTa-Base, too.

**Representation analysis.** Inspired by a recent work that reveals the importance of intermediate layer representation of the LLM backbone [109], we use the last input token embedding of the 24th layer (out of 32 layers in a 7B LLM backbone) for all experiments carried out in the representation space (Figure 1 (b) right, Figure 8, Figure 13, and the JSD computation in Table 4).

# B   Additional Results

## B.1   Ablation Study

We first investigate two important hyperparameters for `Vittle`: (1) bottleneck layer index $l$ and (2) KLD regularization strength $\beta$, where we determined those parameter values based on the average performance on the clean POPE and LB-COCO, while not observing performance on perturbation datasets for fair model selection. We then further explore the impact of the interpolation coefficient $\alpha$, which plays a role in controlling the balance between the original representation and the bottleneck representation. Note that we could not conduct such an extensive search due to the computational burden of training 7B 13B scale models, so the hyperparameter values found here may not be optimal, and `Vittle` can achieve better results with further hyperparameter tuning.

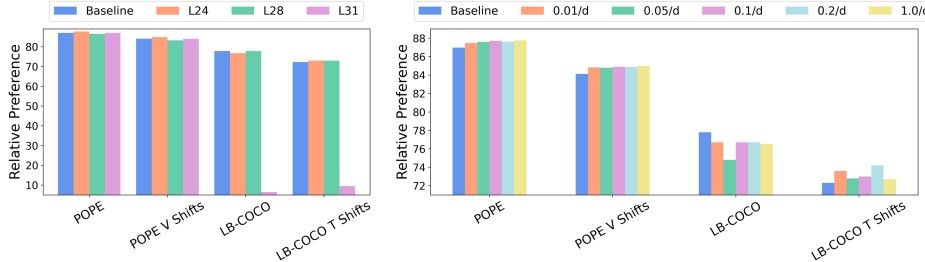

Figure 12: **Ablation study for the bottleneck layer index (left) and KLD regularization magnitude parameter $\beta$ (right).**

For bottleneck target layer ablation (Figure 12 left), we swept across {8, 16, 20, 24, 28, 31} out of 32 layers of the 7B-size LLM backbone. However, applying the bottleneck on the early layer failed to make the language modeling loss converge, so we only provided results for 24, 28, and 31 layers. We observed that intermediate layers (L24 and L28) achieve better results than the penultimate layer (L31), and L24 shows better results on POPE while L28 outperforms L24 on LB-COCO. In conclusion, applying the bottleneck to too early parts hinders shaping some shallow syntactic features that will be actively used at later parts of the layers [110], whereas applying it to too late parts hurts output-specific alignment or formatting [111], which guide us to decide intermediate layer, i.e., 24th,

as a default choice. This is in line with a recent finding that the intermediate layer of LLM matters more than the early or later layers by showing that the quality measurements of the intermediate layer representations have a stronger correlation with performance in downstream tasks [109]. Although we can search different layer indices for visual and textual tokens, we leave this to future work.

For KLD regularization strength parameter ablation (Figure 12 right), we swept across $\{0.01, 0.05, 0.1, 0.2, 1.0\}$, and found that in the POPE dataset, strong regularization results in better performance, whereas it is not the case for LB-COCO. We choose $0.1/d$ as our default, which induces balanced clean-data performance on these two tasks.

Table 11: **Ablation study for the representation interpolation coefficient $\alpha$ of the bottleneck layer.** We observe that using the bottlenecked representation beyond the half portion of the total hinders the convergence of the language modeling loss.

| Alpha | POPE | POPE V Shifts Avg. | LB-COCO | LB-COCO T Shifts Avg. |
|---|---|---|---|---|
| Baseline | 86.98 | 84.12 | 77.8 | 72.3 |
| 0.1 | 87.22 | 84.20 | **77.9** | **73.1** |
| 0.25 | 87.34 | 84.47 | 75.6 | **73.1** |
| 0.5 | **87.71** | **84.90** | 76.7 | 73.0 |
| 0.75 | | | Failed to converge | |
| 1 | | | Failed to converge | |

We also explore the effect of the representation interpolation parameter $\alpha \in [0, 1]$, which can be interpreted as a gating mechanism to control the information flow. As $\alpha$ approaches one, the later parts of the LLM backbone (LLM head in our notation) mainly use the information-penalized representation, while if $\alpha$ becomes smaller, the model strongly relies on the original representation. In Table 11, we observe that using too large values of $\alpha$ results in diverging language modeling loss, indicating that using a strongly penalized representation only cannot predict proper response tokens in sequence. Meanwhile, the larger value of $\alpha$ induces better POPE performance, whereas the trend is inconsistent in the LB-COCO data set, which is consistent with the observations from the previous ablation study in $\beta$.

## B.2 Full Results of Pair-wise Cosine Distance Comparison

We speculate that the performance degradation of MLLMs under perturbations originates from the representation discrepancy between clean and perturbed samples. That is, in the ideal case, a clean sample and its semantically equivalent perturbed sample should be closely mapped in the representation space, but current MLLMs did not shape the representation space in that way (see Figure 1 and Figure 8). In Figure 13, we provide the histograms of representation space pair-wise cosine distance between clean and perturbed examples in 27 types of perturbations. As we can see, `Vittle` (F) consistently mitigates the representation gap by reducing the pair-wise distance over diverse types of perturbations.

## B.3 Full Results with LLaVA-v1.5-7B and LLaVA-v1.5-13B

Table 12 summarizes the overall results of our perturbation benchmarks on object hallucination detection (POPE) and open-ended QA tasks (LB-COCO). We note two findings here: (1) weight-space regularization methods, such as LoRA and WD failed to achieve reasonable performance; (2) although information maximization-based instruction tuning methods, such as ROSS and LIT, somewhat improve performance on POPE and its perturbation datasets, they greatly underperform `Vittle`, indicating a non-trivial challenge to design a versatile instruction tuning objective that can improve MLLMs on broad tasks. Meanwhile, we explore whether `Vittle` can be effective for a much larger model, e.g., a 13B-scale model. Table 13 shows that `Vittle` achieves consistent performance gains in object hallucination detection and open-ended QA tasks under distribution shifts, implying the scalability of our method.

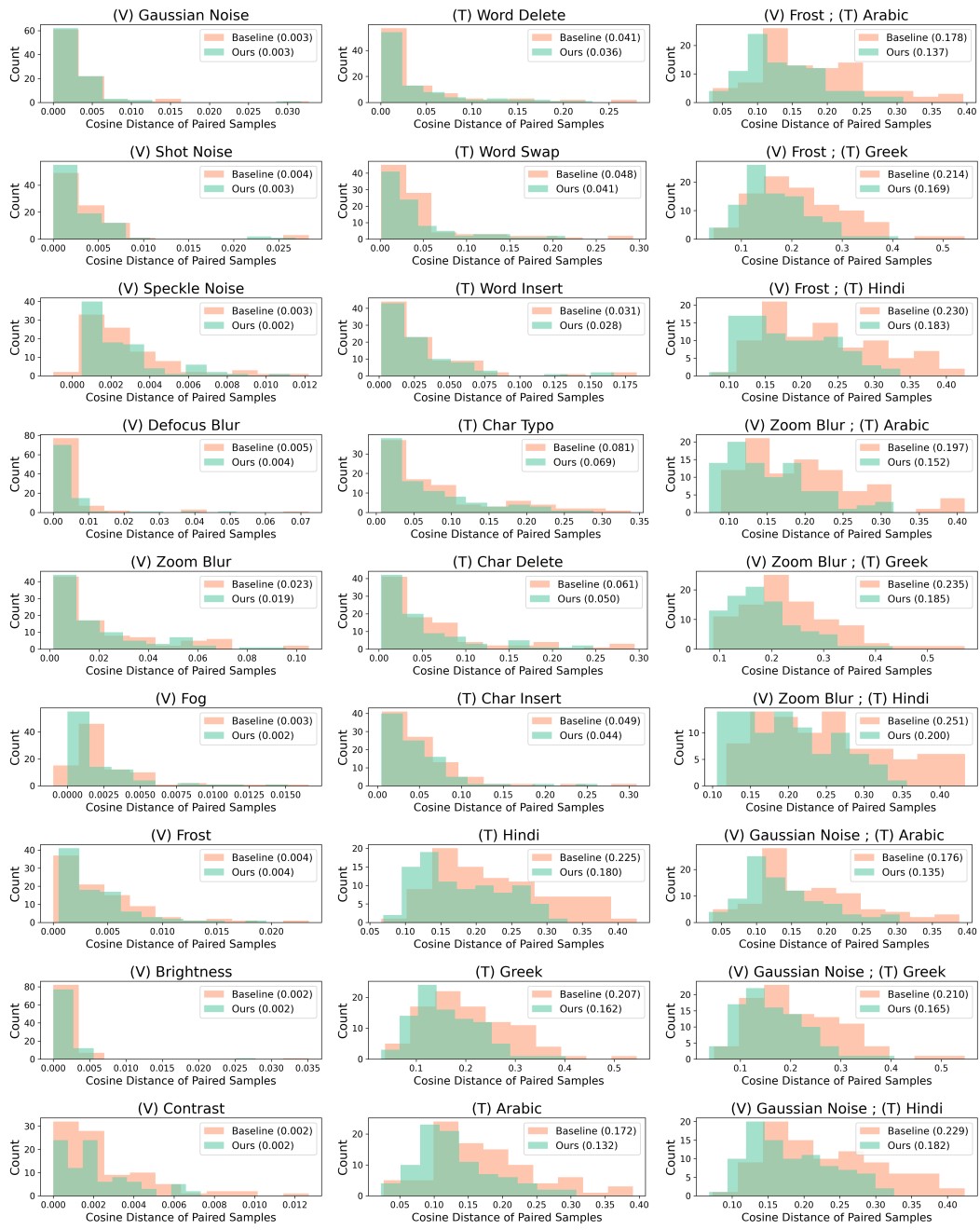

Figure 13: **Pair-wise cosine distance of intermediate representations between clean LB-COCO and 27 versions of perturbed LB-COCO datasets.** `Vittle` (F) consistently reduces the representation gap between the clean samples and their semantically equivalent perturbed ones.

Table 12: **Comparison with alternative training approaches.** We compare `Vittle` with weight-space regularization methods, LoRA [83] and weight decay (WD), and two recent visual instruction tuning learning objectives, ROSS [84] and LIT [85] on LLaVA-v1.5-7B model. Evaluations are conducted on POPE, its nine visually perturbed variants (POPE V Pert.), LB-COCO, and its nine {visually/textually/jointly} perturbed variants, where we mark the best one as bold and the second best one as underlined.

| Method | POPE | POPE V Pert. | LB-COCO | LB-COCO V Pert. | LB-COCO T Pert. | LB-COCO J Pert. |
|---|---|---|---|---|---|---|
| Baseline | 86.98 | 84.12 | **77.8** | 73.4 | 72.2 | 62.3 |
| LoRA | 83.33 | 80.23 | 73.4 | 70.4 | 62.7 | 39.7 |
| WD | 87.22 | 83.97 | 74.1 | 72.1 | 73.0 | 59.5 |
| ROSS | 87.79 | 84.67 | 74.4 | 72.0 | 71.3 | 60.0 |
| LIT | 87.38 | 84.21 | 77.5 | 72.1 | 72.9 | 58.9 |
| `Vittle` (L) | 87.71 | 84.91 | 76.7 | 73.9 | 73.0 | 62.7 |
| `Vittle` (F) | **87.81** | **84.99** | 76.1 | **74.2** | **74.1** | **64.4** |

Table 13: `Vittle` **on LLaVA-v1.5-13B model.** We compare `Vittle` with the standard learning objective on LLaVA-v1.5-13B model that uses Vicuna-v1.5-13B as an LLM backbone. We set the bottleneck layer index $l = 36$, interpolation coefficient $\alpha = 0.5$, and bottleneck KLD regularization strength $\beta = \frac{0.1}{d}$. `Vittle` outperforms baseline on perturbed datasets while showing rivaling performance on the clean dataset.

| Method | POPE | POPE V Pert. | LB-COCO | LB-COCO V Pert. | LB-COCO T Pert. | LB-COCO J Pert. |
|---|---|---|---|---|---|---|
| Baseline | 87.14 | 84.02 | **76.9** | 73.5 | 73.8 | 64.6 |
| `Vittle` (L) | 87.22 | **84.85** | 76.6 | **74.5** | **74.0** | **65.4** |
| `Vittle` (F) | **87.32** | 84.65 | 76.8 | 74.2 | 73.9 | 65.3 |

## B.4 Further Investigation on `Vittle`

**Peak memory allocation.** `Vittle` hosts MLP blocks inside the LLM backbone to model the posterior distributions of the inner representations, and compute additional learning signals, e.g., KLD between priors and posteriors. In Table 5, we reported the wall-clock runtime, and here, we compare the maximum peak memory allocation across GPU chips (8 chips during the training and 1 chip during the inference) in Table 14. We see that the memory overhead induced by `Vittle` is marginal across both training and inference.

Table 14: **Memory comparison.** We compare baseline and `Vittle` (F) in terms of the maximum peak memory allocation (gigabytes; GB) across GPU chips during training and inference.

| Method | Peak Mem GB (train) | Peak Mem GB (test) |
|---|---|---|
| Baseline | 37.55 | 15.62 |
| `Vittle` | 38.98 | 15.84 |

Table 15: **Response diversity comparison.** We report scores of four representative measurements of text diversity on the models' responses to LB-COCO clean data queries.

| Method | Distinct-1 ($\uparrow$) | Distinct-2 ($\uparrow$) | CR ($\downarrow$) | Hom RL ($\downarrow$) |
|---|---|---|---|---|
| Baseline | 0.2356 | 0.6117 | 3.234 | 0.155 |
| `Vittle` (L) | **0.2413** | **0.6279** | **3.212** | 0.147 |
| `Vittle` (F) | 0.2382 | 0.6260 | 3.238 | **0.144** |

**Output diversity comparison.** One may be concerned that learning a compressive representation with `Vittle` can hurt the diversity of the textual output, which is undesirable for a chat assistant. To investigate the output text diversity with and without bottleneck training, we evaluate with four common textual diversity metrics, Distinct n-gram [112], as well as the compression ratio (CR) and

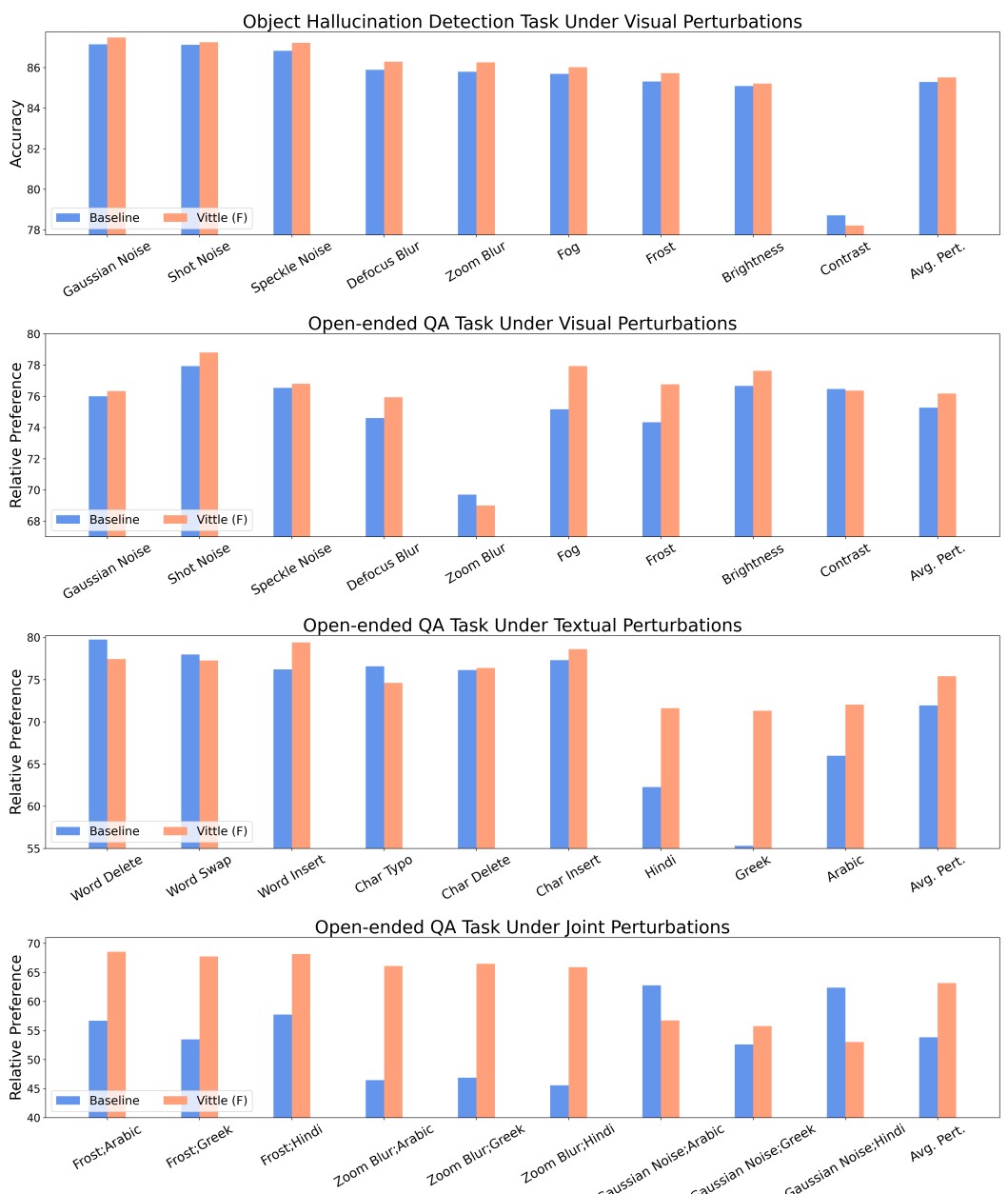

Figure 14: **Object hallucination detection performance on POPE perturbation datasets (top), and Open-ended QA performance on LB-COCO perturbation datasets (three below) of Prism-7B**. We enumerate the accuracy for the object hallucination detection task and relative preference score for the open-ended QA task of each method on perturbed datasets, where we observe consistent performance gains by `Vittle`.

the homogeneity score driven by Rouge-L (Hom RL) [113] over the outputs from each model trained by the baseline method and `Vittle` on the LB-COCO clean dataset.

## B.5 Applicability to Other MLLMs

We now investigate `Vittle`'s effectiveness on another recent MLLM, Prism-7B, beyond LLaVA. As noted in Section A.1, Prism has quite a different design principle than LLaVA with respect to the visual encoder and the training strategy, so it is suitable for investigating the versatility of `Vittle`

between models. Table 16 shows summarized results on our perturbation benchmarks[14]. In object hallucination detection tasks, `Vittle` outperforms the standard cross-entropy only training baseline on clean and perturbed datasets. In open-ended QA tasks, `Vittle` consistently boosts performance in perturbation scenarios with large margins while maintaining performance on the clean dataset. The results of the perturbation-specific performance comparison are provided in Figure 14.

Table 16: `Vittle` **on Prism-7B model.** We compare `Vittle` (F) with the standard learning objective under the Prism-7B model training regime that adopts two visual encoders (DINOv2 and SigLIP) and the single-stage training rather than two-stage training with a single CLIP visual encoder. `Vittle` significantly improves perturbation-robustness compared with a naive learning objective.

| Method | POPE | POPE V Pert. | LB-COCO | LB-COCO V Pert. | LB-COCO T Pert. | LB-COCO J Pert. |
|---|---|---|---|---|---|---|
| Baseline | 87.54 | 85.29 | 79.4 | 75.3 | 71.9 | 53.8 |
| `Vittle` (F) | 88.11 | 85.52 | 79.0 | 76.2 | 75.4 | 63.2 |

## C  Extended Literature Review

**Compression for generalization.**    There is a rich history in the machine learning field that connects compression of the model or its inner representation to generalization [114], from the classical learning theory with *Occam's razor* [115, 116] and *Minimal Description Length* [117, 118, 119] to *IB principle* [46, 47], by suggesting models that provide minimal and simplest representation of data generalize better [33, 118, 52, 120] analogy to human perception [13, 121, 15]. Recently, Wilson [122] proposed a new generalization bound for contemporary large-scale models where the ***compressibility*** of a learning algorithm plays a key role in better generalization. According to that discussion, even the maximally flexible billion-scale model can have a small effective dimensionality (indicating the higher compressibility) by embracing *soft inductive biases* [123], such as, a regularization term, to the learning problem. On top of these, IB-objective of `Vittle` can be understood as a soft inductive bias to seek a minimal sufficient representation that helps generalization for the challenging queries.

**Robustness of fine-tuned foundation models.**    Although large-scale pre-trained models have appealing generalization capability across diverse data instances from different domains, their fine-tuned counterparts usually hurts that strong generalization capability while being adapting on task-specific in-distribution samples [124, 125]. This undesirable performance compromise between adaptation to in-distribution samples and generalization to samples from broad domains has spurred the community to work on *robust fine-tuning* of foundation models [124, 125, 126, 127, 128, 129, 130]. This line of work addresses the adaptation-robustness trade-off by (1) introducing a regularization term [128, 129], (2) tweaking the training procedure [124, 126], or (3) merging multiple models in the weight space [125, 131]. However, almost all of the existing robust fine-tuning literature has focused on a discriminative model, such as CLIP [71], under classification setups. Although there are a few works on robust instruction tuning of MLLMs [132, 133], they do not specifically focus on improving robustness under diverse types of distribution shifts and propose a *data-centric approach*, i.e., expanding instruction tuning datasets in terms of quantity or diversity, that requires external MLLM-based data generation process and/or careful post-processing from humans. In this work, we take a *representation-centric* approach that modifies the learning objective of visual instruction tuning to efficiently enhance the robustness of MLLM under diverse distribution shifts (27 types in total).

---

[14]Due to resource constraints, we only explore `Vittle` (F) one of our prior distribution instantiations.

# D  Derivation of Variational Bound for IB in MLLM

Here we provide a full derivation for the variational lower bound for IB. The derivation skeleton was mainly inspired by existing works [134, 32]. We begin with the mutual information term $I(Z, X)$. Given the sequential nature of MLLM, we decompose both the input $X = (X_v, X_t)$ and the latent representation $Z = (Z_v, Z_t)$ into **v**isual and **t**extual components. We can then derive the following upper bound for $I(Z, X)$:

$$
\begin{aligned}
I(Z, X) &= \int p(x, z) \log \frac{p(x, z)}{p(x)p(z)} dx dz = \int p(x, z) \log \frac{p(z|x)}{p(z)} dx dz \\
&= \int p(x, z) \log \frac{p(z|x)}{r(z)} dx dz - D_{\mathrm{KL}}(p(z)||r(z)) \\
&\leq \int p(x, z) \log \frac{p(z|x)}{r(z)} dx dz \\
&= \int p(x_v, x_t, z_v, z_t) \log \frac{p(z_t|x_v, x_t) p(z_v|x_v)}{r(z_v) r(z_t)} dx_v dx_t dz_v dz_t \\
&= \int p(x_v, x_t) \int p(z_t|x_v, x_t) \int p(z_v|x_v) \log \frac{p(z_v|x_v)}{r(z_v)} dx_v dx_t dz_v dz_t \\
&\quad + \int p(x_v, x_t) \int p(z_v|x_v) \int p(z_t|x_v, x_t) \log \frac{p(z_t|x_v, x_t)}{r(z_t)} dx_v dx_t dz_v dz_t \\
&= \mathbb{E}_{x_v}[D_{\mathrm{KL}}(p(z_v|x_v)||r(z_v))] + \mathbb{E}_{x_v, x_t}[D_{\mathrm{KL}}(p(z_t|x_v, x_t)||r(z_t))], \quad (9)
\end{aligned}
$$

where the first inequality holds given the non-negativity of $D_{\mathrm{KL}}[r(z), p(z)]$ and $p(z_v|x_v, x_t) = p(z_v|x_v)$ due to causal attention in MLLM. Here, we introduce $r(z) = r(z_v, z_t) = r(z_v)r(z_t)$ as a factorizable variational approximation of the true prior for the latent representation $p(z)$. Next, for the output-relevant term $I(Z, Y)$, we have the lower bound:

$$
\begin{aligned}
I(Z, Y) &= \int p(y, z) \log \frac{p(y, z)}{p(y)p(z)} dy dz = \int p(y, z) \log \frac{p(y|z)}{p(y)} dy dz \\
&= \int p(y, z) \log q(y|z) dy dz + D_{\mathrm{KL}}(p(y|z)||q(y|z)) - \int p(y) \log p(y) dy \\
&\geq \int p(y, z) \log q(y|z) dy dz \\
&= \int p(x, y, z) \log q(y|z) dx dy dz = \int p(x)p(y|x)p(z|x) \log q(y|z) dx dy dz, \\
&= \mathbb{E}_{x, y} \mathbb{E}_{z|x} \left[ \log q(y|z) \right]. \quad (10)
\end{aligned}
$$

where $p(x, y, z) = p(x)p(z|x)p(y|x)$ given the Markov assumption $Y \leftrightarrow X \leftrightarrow Z$, and $p(z|x, y) = p(z|x)$ holds given that the representation $Z$ can not directly depend on $Y$, and the entropy term of $y$, i.e., $\int -p(y) \log p(y) dy = H(Y)$, is ruled out due to its independence for optimization problem. Here, we replace the intractable $p(y|z)$ with a variational approximation $q(y|z)$ that will be parameterized by a model. Finally, combining the lower bound of $I(Z, Y)$ and the upper bound of $I(Z, X)$ yields a variational lower bound for the IB objective as follows,

$$
\begin{aligned}
\mathrm{IB}(X, Y) \geq\ & \mathbb{E}_{x, y} \left[ \mathbb{E}_{z|x}[\log q(y|z)] \right] \\
& - \beta \left( \mathbb{E}_{x_v} \left[ D_{\mathrm{KL}}(p(z_v|x_v)||r(z_v)) \right] + \mathbb{E}_{x_v, x_t} \left[ D_{\mathrm{KL}}(p(z_t|x_v, x_t)||r(z_t)) \right] \right). \quad (11)
\end{aligned}
$$

**Asymmetric posterior $p(z|x)$ modeling in practice.**  As we fed the (pre/post-bottleneck) representation interpolation $\hat{z} = (1 - \alpha)z + \alpha\tilde{z}$ into the LLM-head $f_{\theta^{l+}}$ for output decoding $q(y|\hat{z})$, this yields an asymmetric specification of $p(z|x)$ being modeled in $I(Z, X)$ and $I(Z, Y)$. To be precise, given our posterior instantiation $p(z|x) := \mathcal{N}(z; \mu(x), \sigma^2(x) \cdot I)$ in the upper bound of $I(Z, X)$, which is modeled by the LLM-stem $f_{\theta^l}$ plus the bottleneck layer $g_\phi$, we use $\mathcal{N}(z; \alpha\mu(x) + (1 - \alpha)c, \alpha^2\sigma^2(x) \cdot I)$, where $c = f_{\theta^l}(x)$ denotes a constant vector, to model the $p(z|x)$ in the lower bound of $I(Z, Y)$.

# E  Missing Proof

## E.1  Preliminary

We start by providing a definition of Mutual Information (MI) below.

**Definition E.1** (**Mutual Information (MI)**). *For a joint distribution $P_{XY}$ over $\mathcal{X} \times \mathcal{Y}$, the mutual information with respect to $P_{XY}$ is defined as,*

$$I(P_{XY}) := \mathbb{E}_{x,y \sim P_{XY}}[\log \frac{P_{XY}(x,y)}{P_X(x)P_Y(y)}]. \tag{12}$$

If $X$ is an instruction and $Y$ is a corresponding response, we regard $I(P_{XY})$ as a relevance between the instruction and the response that can be seen as a possible quantification of *instruction following* capability of MLLMs. Effective MI is defined based on the MI as follows:

**Definition E.2** (**Effective Mutual Information (EMI) [19]**). *Given the joint distribution $P_{XY}$ and MLLM $P_\Theta$ parameterized with $\Theta$, the effective mutual information between the input and model response is defined as,*

$$EMI(P_{XY}; P_\Theta) := I(P_{XY_\Theta}) - I(P_{XY}), \tag{13}$$

where $P_{XY_\Theta}$ denotes the joint distribution between the input $X$ and the output of the model $Y_\Theta$. Although the vanilla MI can also be used as a metric to evaluate models' output response by $I(P_{XY_\Theta})$, the scale of it varies depending on the target data distribution which is undesired when our interest is to compare performance of model across multiple domains which can be addressed by EMI. Recall that we are ultimately interested in the performance difference of MLLMs across two different datasets, and this can be captured by the EMI difference (EMID) as follows:

**Definition E.3** (**EMID**). *Let $P_\Theta : \mathcal{X} \to \mathcal{Y}$ be an MLLM with parameters $\Theta$ that produces an output response $Y_\Theta$ given an input instruction $X$. For joint distributions $P_{XY}$ and $Q_{XY}$, effective mutual information difference of $P_\Theta$ over $P$ and $Q$ is defined as below,*

$$EMID(P_{XY}, Q_{XY}; P_\Theta) := [I(P_{XY_\Theta}) - I(P_{XY})] - [I(Q_{XY_\Theta}) - I(Q_{XY})]. \tag{14}$$

By setting $P$ as an instruction tuning distribution (training data) and $Q$ as an arbitrary test time distribution (evaluation data), we prefer a model that has a smaller EMID value, which indicates better robustness under distribution shifts between $P$ and $Q$. Now, based on the original theorem provided by Oh et al. [19], we are ready to derive a new upper bound for EMID tailored to our representation-centric visual instruction tuning setup.

## E.2  A New Upper Bound for Effective Mutual Information Difference

We first review Lemma 1 of Shui et al. [135] and its adapted version, a conditional entropy bound [19] as follows,

**Lemma E.4** (Lemma 1 from Shui et al. [135]). *Let $Z \in \mathcal{Z}$ be the real-valued integrable random variable, and denoting two distributions on a common space $\mathcal{Z}$ by $P$ and $Q$ such that $Q$ is absolutely continuous w.r.t. $P$. If for any function $f$ and $\lambda \in \mathbb{R}$ such that $\mathbb{E}_P[\exp(\lambda(f(z) - \mathbb{E}_P(f(z))))] < \infty$, then we have:*

$$\lambda(\mathbb{E}_{z \sim Q}[f(z)] - \mathbb{E}_{z \sim P}[f(z)]) \leq D_{\mathrm{KL}}(Q||P)$$
$$+ \log \mathbb{E}_{z \sim P}[\exp(\lambda(f(z) - \mathbb{E}_{z \sim P}[f(z)]))]$$

**Lemma E.5** (Conditional entropy bound [19]). *Let $f(x) := H(Q_{Y|x})$ and $\hat{H}(Q_{Y|x}) := \max_{x \in \mathcal{X}} H(Q_{Y|x})$, given the marginal distributions $P_X$ and $Q_X$, and conditional distributions $P_{Y|X}$ and $Q_{Y|X}$, according to Lemma E.4, we have a conditional upper bound:*

$$i) \quad \mathbb{E}_{x \sim P}[H(Q_{Y|x})] - \mathbb{E}_{x \sim Q}[H(Q_{Y|x})] \leq \hat{H}(Q_{Y|\mathbf{x}})\sqrt{2D_{\mathrm{JS}}(P_X||Q_X)}.$$

*Similarly, given the marginal distribution $P_X$ and $Q_X$, and an MLLM $P_\Theta$, let $f(x) := H(P_\Theta(\cdot|x))$ and $\hat{H}(P_\Theta) := \max_{x \in \mathcal{X}} H(P_\Theta(\cdot|x))$, then, according to Lemma E.4, we have another conditional upper bound:*

$$ii) \quad \mathbb{E}_{x \sim Q}[H(P_\Theta(\cdot|x)] - \mathbb{E}_{x \sim P}[H(P_\Theta(\cdot|x))] \leq \hat{H}(P_\Theta)\sqrt{2D_{\mathrm{JS}}(P_X||Q_X)}.$$

Next, we should also need to formulate the relationship between JSD in the input space and JSD in the representation space, which is done through Lemma E.6.

**Lemma E.6.** *Let $f : \mathcal{X} \to \mathcal{Z}$ be an encoder that maps an input $X$ to a representation $Z$, for the input distributions $P_X$ and $Q_X$ and $f$-induced representation distribution $P_Z$ and $Q_Z$, we have an inequality below,*

$$\sqrt{2D_{\mathrm{JS}}(P_X||Q_X)} \leq \sqrt{2D_{\mathrm{JS}}(P_Z||Q_Z)} + \sqrt{\mathbb{E}_{z\sim P}[D_{\mathrm{KL}}(P_{X|z}||M_{X|z})] + \mathbb{E}_{z\sim Q}[D_{\mathrm{KL}}(Q_{X|z}||M_{X|z})]} \tag{15}$$

where $M_{X|z} := \frac{P_{X|z} + Q_{X|z}}{2}$.

*Proof.* We start from the definition of JSD,

$$D_{\mathrm{JS}}(P_X||Q_X) = \frac{1}{2}D_{\mathrm{KL}}(P_X||M_X) + \frac{1}{2}D_{\mathrm{KL}}(Q_X||M_X), \quad M_X = \frac{P_X + Q_X}{2}.$$

By applying the chain rule of KLD under a deterministic map[15] $X \to Z$, we know that,

$$D_{\mathrm{KL}}(P_{XZ}||M_{XZ}) = D_{\mathrm{KL}}(P_Z||M_Z) + \int P_Z(z)D_{\mathrm{KL}}(P_{X|z}||M_{X|z})dz$$

$$= D_{\mathrm{KL}}(P_X||M_X) + \int P_X(x)\cancel{D_{\mathrm{KL}}(P_{Z|x}||M_{Z|x})}dx$$

$$\Leftrightarrow D_{\mathrm{KL}}(P_X||M_X)$$

Then, we have,

$$D_{\mathrm{JS}}(P_X||Q_X) = D_{\mathrm{JS}}(P_Z||Q_Z) + \frac{1}{2}(\mathbb{E}_{z\sim P}[D_{\mathrm{KL}}(P_{X|z}||M_{X|z})] + \mathbb{E}_{z\sim Q}[D_{\mathrm{KL}}(Q_{X|z}||M_{X|z})]),$$

which results in ineq. (15) by applying the triangular inequality after multiplying 2 on both sides. $\square$

Now we derive a new upper bound for EMID, which is defined over the representation space rather than the previous one defined over the input space [19] in Proposition E.7.

**Proposition E.7** (**EMID upper bound**). *Let $P_\Theta$ be an MLLM that maps $X = \{X_v, X_t\}$ to $Z = \{Z_v, Z_\Theta\}$, and then subsequently maps $Z$ to $Y_\Theta$. Given joint distributions $P_{XY} = P_X \times P_{Y|X}$ and $Q_{XY} = Q_X \times Q_{Y|X}$, by assuming consistent conditional distributions over $Z_v|Z_t$, $Z_t|Z_v$, and $Y|X$ between $P$ and $Q$, we have an upper bound for $EMID(P_{XY}, Q_{XY}; P_\Theta)$ as follow,*

$$\hat{H}\left(D_{\mathrm{JS}}^{\frac{1}{2}}(P_{Z_v}||Q_{Z_v}) + D_{\mathrm{JS}}^{\frac{1}{2}}(P_{Z_t}||Q_{Z_t}) + \sqrt{\Delta_{X|Z}}\right) + |H(P_{Y_\Theta}) - H(P_Y)| + |H(Q_{Y_\Theta}) - H(Q_Y)|, \tag{16}$$

where $H$ and $D_{\mathrm{JS}}^{\frac{1}{2}}$ indicate the entropy and square root of Jensen-Shannon divergence (JSD), respectively, $\Delta_{X|Z} := \mathbb{E}_{z\sim P}[D_{\mathrm{KL}}(P_{X|z}||M_{X|z})] + \mathbb{E}_{z\sim Q}[D_{\mathrm{KL}}(Q_{X|z}||M_{X|z})]$ with a mixture distribution $M = \frac{P+Q}{2}$, and $\hat{H} := \max_{x\in\mathcal{X}}[H(Q_{Y|x}) + H(P_{Y_\Theta})]$.

*Proof.* Given the entropy-based definition of the mutual information, $I(P_{XY}) := H(P_Y) - \mathbb{E}_{x\sim P}[H(P_{Y|x})]$, let $P_{Y_\Theta} = \mathbb{E}_{x\sim P}[P_\Theta(\cdot|x)]$ and $Q_{Y_\Theta} = \mathbb{E}_{x\sim Q}[P_\Theta(\cdot|x)]$, then, EMID can be expressed as follows,

$$\begin{aligned}
&\mathrm{EMID}(P_{XY}, Q_{XY}; P_\Theta) \\
&= \mathrm{EMI}(P_{XY}; P_\Theta) - \mathrm{EMI}(Q_{XY}; P_\Theta) \\
&= (H(P_{Y_\Theta}) - \mathbb{E}_{x\sim P}[H(P_\Theta(\cdot|x))] - H(P_Y) + H(P_{Y|X})) \\
&\quad - (H(Q_{Y_\Theta}) - \mathbb{E}_{x\sim Q}[H(P_\Theta(\cdot|x))] - H(Q_Y) + H(Q_{Y|X})) \\
&\leq (H(P_{Y|X}) - H(Q_{Y|X})) + (\mathbb{E}_{x\sim Q}[H(P_\Theta(\cdot|x))] - \mathbb{E}_{x\sim P}[H(P_\Theta(\cdot|x))]) \\
&\quad + |H(P_{Y_\Theta}) - H(P_Y) + H(Q_Y) - H(Q_{Y_\Theta})| \\
&\leq \underbrace{(H(P_{Y|X}) - H(Q_{Y|X}))}_{(A)} + \underbrace{(\mathbb{E}_{x\sim Q}[H(P_\Theta(\cdot|x))] - \mathbb{E}_{x\sim P}[H(P_\Theta(\cdot|x))])}_{(B)} \\
&\quad + |H(P_{Y_\Theta}) - H(P_Y)| + |H(Q_Y) - H(Q_{Y_\Theta})|. \tag{17}
\end{aligned}$$

---

[15]Note that although `Vittle` governs a probabilistic encoder during training, it produces deterministic output during inference by using the learned posterior mean rather than a random sample (Section 3.3).

Moreover, we have the following inequality for $H(P_{Y|X})$ proposed by [19],

$$H(P_{Y|X}) - H(Q_{Y|X}) \le 4\mathbb{E}_{x \sim P}[D_{\mathrm{JS}}^{\frac{1}{4}}(P_{Y|x}||Q_{Y|x})] + \mathbb{E}_{x \sim P}[H(Q_{Y|x})] + \mathbb{E}_{x \sim Q}[H(Q_{Y|x})] \quad (18)$$

By plugging inequalities in Lemma E.5 and ineq. (18) into the ineq. (17) to replace the terms (A) and (B), and given the consistent conditional distribution assumption for $Y|X$, i.e., $P_{Y|X} = Q_{Y|X}$, we have a much simpler upper bound as follows,

$$\mathrm{EMID}(P_{XY}, Q_{XY}; P_\Theta) \le \hat{H}\sqrt{2D_{\mathrm{JS}}(P_X||Q_X)} + |H(P_{Y_\Theta}) - H(P_Y)| + |H(Q_Y) - H(Q_{Y_\Theta})|,$$

where $\hat{H} := \max_{x \in \mathcal{X}}[H(Q_{Y|x}) + H(P_{Y_\Theta})]$. Then, we can further replace the term $D_{\mathrm{JS}}(P_X||Q_X)$ by using Lemma E.6 to get a bound defined by representation divergence as below,

$$\mathrm{EMID}(P_{XY}, Q_{XY}; P_\Theta)$$
$$\le \hat{H}(\sqrt{2D_{\mathrm{JS}}(P_Z||Q_Z)} + \sqrt{\mathbb{E}_{z \sim P}[D_{\mathrm{KL}}(P_{X|z}||M_{X|z})] + \mathbb{E}_{z \sim Q}[D_{\mathrm{KL}}(Q_{X|z}||M_{X|z})]})$$
$$+ |H(P_{Y_\Theta}) - H(P_Y)| + |H(Q_Y) - H(Q_{Y_\Theta})|. \quad (19)$$

Meanwhile, the chain rule of KLD and the definition of JSD with our consistency assumption for conditional distributions for $Z_v|Z_t$ and $Z_t|Z_v$, one can easily show below.

$$2D_{\mathrm{JS}}(P_{Z_v Z_t}||Q_{Z_v Z_t}) = D_{\mathrm{KL}}(P_{Z_v Z_t}||M_{Z_v Z_t}) + D_{\mathrm{KL}}(Q_{Z_v Z_t}||M_{Z_v Z_t})$$
$$= D_{\mathrm{JS}}(P_{Z_v}||Q_{Z_v}) + D_{\mathrm{JS}}(P_{Z_t}||Q_{Z_t}) \quad (20)$$

Plugging Eq. 20 into ineq. (19) and applying the triangular inequality complete the proof. $\qquad \square$

# F   Impact Statement

Multimodal large language models (MLLMs) today have many societal applications. This work tackles the robustness of MLLMs to distribution shifts between training and test time data. We observed a consistent improvement of our proposal `Vittle` in various types of visual and textual shifts, allowing users to trust the model more than before to safely use AI in a variety of environments. Moreover, although we focused on the robustness perspective in this work, improved invariance-sensitivity trade-off also benefits the fairness-discriminativeness trade-off, which is another crucial desideratum towards reliable AI. Meanwhile, even though its robustness to distribution shifts was improved, there are still potential misuse cases with MLLMs that can affect humanity by producing systematically biased outputs, given the existence of some adversarial data providers or attackers.

