# OpenReview forum: "Visual Instruction Bottleneck Tuning"
_NeurIPS.cc/2025/Conference — NeurIPS 2025 poster_

### Official Review · Reviewer_BHRh · 2025-06-26

**Clarity:** 3
**Significance:** 2
**Originality:** 2
**Rating:** 5
**Confidence:** 3

**Summary:**

This paper investigates the enhancement of MLLM adaptability to perturbations by leveraging the information bottleneck concept. A bottleneck layer is inserted into LLaVA's LLM output layers. The authors provide theoretical grounding to demonstrate that adding this layer corresponds to the theoretical lower bound of the information bottleneck. Furthermore, experiments are conducted on 3 models (2 different-sized LLaVA and Prism-7B) to evaluate and enhance the robustness of MLLMs against perturbations.

**Questions:**

1. The article emphasizes the concept of IB, but its implementation closely resembles VIB normalization, with similar work also seen in CIB [A]. Could you highlight the innovation of this article in comparison to such VIB normalization methods?

2. The bottleneck layer proposed in the article is designed to enhance robustness against input perturbations, as evidenced by the experimental results. How complementary are these benefits to common data augmentation schemes?

3.The experiments in the article are primarily conducted on two - stream VLM. To increase the credibility of the results, could you provide additional experimental results on one - stream VLM?

**Ethical Concerns:**

["NO or VERY MINOR ethics concerns only"]

**Final Justification:**

Q1: Theoretical distinction between CIB and Vittle: The authors' detailed clarification sufficiently explains the theoretical and operational differences between CIB and the proposed Vittle framework.

Q2: Complementarity with Augmentation: The supplemental ablation studies have empirically demonstrated partial complementarity with standard augmentation techniques. This evidence is acceptable.

Q3:  About the application on different VLM architectures: The results on the 2-stream structured VLM are also quite convincing.


According to the above considerations,  I raise the score to 5.

**Limitations:**

yes

**Quality:**

3

**Strengths And Weaknesses:**

Strengths:
1. The authors provide a theoretical justification for Vittle by connecting it to an information-theoretic robustness metric of MLLMs;
2. The method proposed in this paper is simple and applicable for further research and application;
3. The paper is well written.

Weaknesses:
1. The article emphasizes the concept of IB, but its implementation closely resembles VIB normalization, with similar work also seen in CIB [A]. Could you highlight the innovation of this article in comparison to such VIB normalization methods?

2. The bottleneck layer proposed in the article is designed to enhance robustness against input perturbations, as evidenced by the experimental results. How complementary are these benefits to common data augmentation schemes?

3.The experiments in the article are primarily conducted on two - stream VLM. To increase the credibility of the results, could you provide additional experimental results on one - stream VLM?

[A]  Correlation information bottleneck: Towards adapting pretrained multimodal models for robust visual question answering. International Journal of Computer Vision, 132(1), 185–207.

---

> ### Author Rebuttal · Authors · 2025-07-30
>
> Thank you for taking your time to give us the professional comments; We would be happy to answer your questions (Q) below.
>
> > Q1: The article emphasizes the concept of IB, but its implementation closely resembles VIB normalization, with similar work also seen in CIB. Could you highlight the innovation of this article in comparison to such VIB normalization methods?
>
> Thank you for pointing out this relevant work, which we had missed. We agree that CIB [Jiang et al., 2023] is relevant and will cite it in the next revision. While both approaches are inspired by the information bottleneck principle, our contributions differ significantly in terms of objective formulation, implementation, and experimental scope:
>
> 1.  **Objective formulation**: Although both CIB and ours use the variational lower bound for the mutual information $I(Z,Y)$, CIB adopted a multimodal-correlation driven upper bound for $I(Z,X)$, whereas we derive an autoregressive conditional language modeling-specific upper bound for $I(Z,X)$ with variational inference. **This results in distinct formulations tailored to sequence‑generation‑based instruction tuning** (See Eq. (6) of CIB paper and Eq. (4) of our paper).
> 2. **Implementation**: CIB estimates $I(Z,X)$ with external neural estimators (e.g., CLUB, NWJ), which can introduce instability. By contrast, Vittle uses a closed‑form KL divergence formulation, avoiding the variance of estimator‑based methods. In addition, we proposed a novel interpolation between pre‑ and post‑bottleneck representations, where the interpolation coefficient $\alpha$ is progressively increased during training. **This design is unique to our work, and plays a crucial role in achieving stable multimodal instruction tuning** (see Table 10 in the Appendix), as naive adoption of a hard bottleneck ($\alpha$ from the start) leads to unstable training and poor downstream performance.
> 3. **Experimental scope**: CIB evaluated robustness in a _multi‑choice VQA classification setup_ with BERT‑scale bi-directional attention transformer, which differ substantially from today’s autoregressive LLM‑based MLLMs. In contrast, our work applies Vittle to **billion‑scale causally-masked attention transfomer backbones in autoregressive instruction tuning, aligning with modern MLLM design**. This allows us to evaluate on a much broader set of tasks—including **open‑ended VQA**, closed‑form VQA, and hallucination detection, whereas CIB narrowly focused on closed-form VQA tasks. Moreover, our robustness evaluation is extensive, spanning **45 diverse datasets and 30 distribution shifts** across visual, textual, and joint shifts, providing one of the most comprehensive robustness studies to date for multimodal LLMs.
>
> These points highlight the originality and unique contribution of this work, which we believe is a crucial milestone for robust instruction tuning of modern MLLMs that are quite unexplored yet.
>
>
> > Q2: The bottleneck layer proposed in the article is designed to enhance robustness against input perturbations, as evidenced by the experimental results. How complementary are these benefits to common data augmentation schemes?
>
> We thank the reviewer for this thoughtful question. Vittle is complementary to data augmentation: augmentation methods improve robustness by exposing the model to perturbed inputs during training, while Vittle directly regularizes the internal representations via the information bottleneck, making them more invariant to nuisance factors. This difference in mechanism means the two approaches can potentially reinforce each other. In fact, preliminary experiments where we combined Vittle with standard visual augmentations (SimCLR Augmentation [Chen et al. 2020], AutoAugment [Cubuk et al. 2018], and RandAugment [Cubuk et al. 2019]) showed additive gains in robustness over using either approach alone.
>
> | Augmentation       | Method     | POPE V Shift |
> |--------------------|------------|--------------|
> | -         | Baseline   |       83.38       |
> | -         | Vittle (F) |       84.74       |
> | SimCLR Aug         | Baseline   |   85.00           |
> | SimCLR Aug         | Vittle (F) |   85.67           |
> | AutoAugment   | Baseline   |     84.99         |
> | AutoAugment   | Vittle (F) |     85.91         |
> | RandAugment | Baseline   |      84.39        |
> | RandAugment | Vittle (F) |      85.37        |
>
>
>
> > Q3: The experiments in the article are primarily conducted on two - stream VLM. To increase the credibility of the results, could you provide additional experimental results on one - stream VLM?
>
> We would like to clarify that our main experiments are already conducted on one‑stream architectures, such as LLaVA‑v1.5, where visual features are projected into the LLM embedding space and processed jointly with text tokens by the same backbone. According to the definition in a vision-language pre-training survey paper [Chen et al. 2022], LLaVA belongs to the one-stream VLM. This setup is widely adopted in recent multimodal LLM research, ensuring fair comparison with prior work. To further demonstrate generality, we also evaluated Vittle on Prism-7B in Appendix, another strong one‑stream VLM, and observed consistent robustness improvements under distribution shifts.
>
> ### Reference
> - Jiang et al. 2023, Correlation Information Bottleneck: Towards Adapting Pretrained Multimodal Models for Robust Visual Question Answering
> - Cheng et al. 2020, CLUB: A Contrastive Log-ratio Upper Bound of Mutual Information
> - Belghazi et al. 2018, Mutual Information Neural Estimation
> - Cubuk et al. 2018, AutoAugment: Learning Augmentation Policies from Data
> - Cubuk et al. 2019, RandAugment: Practical automated data augmentation with a reduced search space
> - Chen et al. 2020, A Simple Framework for Contrastive Learning of Visual Representations
> - Chen et al. 2022, VLP: A Survey on Vision-Language Pre-training

---

> > ### Comment · Reviewer_BHRh · 2025-08-01
> >
> > Q1: Theoretical distinction between CIB and Vittle
> > The authors' detailed clarification sufficiently explains the theoretical and operational differences between CIB and the proposed Vittle framework.
> >
> > Q2: Complementarity with Augmentation
> > The supplemental ablation studies have empirically demonstrated partial complementarity with standard augmentation techniques. This evidence is acceptable.
> >
> > Q3: Architectural consideration
> > Apologize for my error in the initial inquiry.  To clarify:  The experiments in the article are primarily conducted on one-stream VLM. To increase the credibility of the results, could you provide additional experimental results on two-stream VLM? Thank you.

---

> ### Author Response · Authors · 2025-08-05
> **Follow-up Response to BHRh**
>
> We appreciate Reviewer BHRh's active discussion and questioning on broad applicability of our method. After release of LLaVA-style single-stream VLMs (multimodal LLMs we have referred in this paper) have became a de-facto standard architecture of multimodal instruction-following models, so we focused on LLaVA-style single-stream model through our entire draft.
>
> However, it would be also worhty to test how our Vittle can be applied on diverse types of architectures in general. For this, we take into account **BEiT-3 model** [Wang et al. 2023] (dual-stream VLM according to the Table 2 of CIB paper [Jiang et al. 2024]) fine-tuning on COCO image captioning setup by following the training configurations of the Miscrosoft official codebase -- unlim/beit3.
>
> > Setup
> * We train `beit3_base_patch16_224` backbone model for 5 epochs on COCO 2014 dataset of karphathy split without data augmentation, which is slightly different to the authors' proposed fine-tuning setup (10 epoch training of beit3_base_patch16_480 with data augmentation) due to time and resource constraint.
> * We compared the baseline BEiT-3 fine-tuning method and our bottleneck-applied method on clean COCO images and perterbed COCO images (nine perturbations considered in our original draft) in terms of five standard captioning metrics. We insert the bottleneck layer on top of the final layer of multiway transformer block stacks, assuming fixed isotropic Guaaisan prior, and set the parameters $\alpha$ and $\beta$ to 0.5 and 1.0, respectively.
>
> | Method   | Data      | Bleu_4 | METEOR | ROUGE_L | CIDEr | SPICE |
> |----------|-----------|--------|--------|---------|-------|-------|
> | Baseline | Clean     | 0.373  | 0.295  | 0.585   | 1.267 | 0.229 |
> | Bottleneck     | Clean     | **0.386**  | **0.303**  | **0.594**   | **1.303** | **0.236** |
> | Baseline | Nine Pert. Avg. | 0.347  | 0.280  | 0.565   | 1.159 | 0.214 |
> | Bottleneck     | Nine Pert. Avg. | **0.350**  | **0.284**  | **0.568**   | **1.174** | **0.216** |
>
> We observe that bottleneck-applied fine-tuning induces consistently better performance across all the considered metrics both in clean and perturbed image data settings. It is also worth to noting that we could not extensively tune the hyperparmeter (bottleneck insertion layer, $\alpha$, and $\beta$) due to limited time. This demonstrates the generality of our bottleneck-based fine-tuning approach which can be robustly applied to different types of model architectures.
>
> Is this result enough to address your concern/question? If you have any other opinions or questions, please let us know. We would be happy to discuss further with you.
>
> ---
>
> ### Reference
> - Wang et al. 2023, Image as a Foreign Language: BEIT Pretraining for Vision and Vision-Language Tasks
> - Jiang et al. 2024, Correlation Information Bottleneck: Towards Adapting Pretrained Multimodal Models for Robust Visual Question Answering

---

> > ### Comment · Reviewer_BHRh · 2025-08-06
> >
> > Thank the authors for the detailed experiments in such a short time. The results on the 2-stream structured VLM are also quite convincing. I will raise the score to 5.

---

> > > ### Author Response · Authors · 2025-08-06
> > >
> > > Dear Reviewer BHRh,
> > >
> > > We would like to express our sincere gratitude for your professional comments that guide us to further explore some important details and working conditions of our methods. We are also very happy to note that you found a couple of strengths in our paper, i.e., theoretical justification, simplicity, and broad applicability of our method, and presentation quality. Besides, we believe that our manuscript will be significantly enhanced by reflecting your feedback, e.g., incorporating the additional results and adding missing literature (CIB paper).
> > >
> > >
> > > Thanks for taking your time to review our paper, and reconsidering your rating towards clearer acceptance!
> > >
> > > Best regards,
> > > The authors

---

### Official Review · Reviewer_haEx · 2025-06-30

**Clarity:** 3
**Significance:** 2
**Originality:** 3
**Rating:** 4
**Confidence:** 3

**Summary:**

The paper proposes Visual Instruction Bottleneck Tuning (Vittle) to enhance the robustness of MLLMs under distribution shifts. The technique is built upon the information-bottleneck theory and specially designed for resolving the distribution shift problem of MLLMs.

**Questions:**

See Weaknesses.

**Ethical Concerns:**

["NO or VERY MINOR ethics concerns only"]

**Final Justification:**

The authors have answered my question. My only concern of this paper is the limited performance improvement. I thus keep the score of "Borderline accept".

**Limitations:**

See the third question in Weaknesses.

**Quality:**

3

**Strengths And Weaknesses:**

Strengths:
This paper proposes a theoretically sound technique to resolve the distribution shift problem for MLLMs. The experimental results show the  effectiveness of the proposed method. The implementation is clear and can easily be reproduced.

Weaknesses:
1. It looks the performance of Vittle(F) and Vittle(L) varies across tasks. For example, Vittle(L) excels in long-tail scenarios, while Vittle(F) performs better under perturbations. Since in many benchmarks either Vittle(F) or Vittle(L) brings very limited or even negative performance gain (e.g., Vittle(F) in Table 2 and Vittle(L) in Figure 5 left), it seems Vittle requires task-specific hyper-param choice (F or L) under distribution shift.
2. The relationship between IB and model robustness in distribution shift is not clearly stated. Why "This property is particularly
129 desirable for robust instruction tuning"(Line 128-129)? Can IB enhance the robustness of other models in other tasks (e.g., CNNs in image classification) under distribution shift?
3. Vittle introduces 20% extra training time and 1.5% extra parameters compared to baseline. Concerning that in many benchmarks Vittle brings very limited performance gain, is it possible that the difference simply comes from the increase of the training time and parameters? I think it is necessary for the authors to clarify this problem.

---

> ### Author Rebuttal · Authors · 2025-07-30
>
> We appreciate your productive comments and feedback, and happy to provide our response to the weaknesses (W) identified by you.
>
> > W1: It looks the performance of Vittle(F) and Vittle(L) varies across tasks. For example, Vittle(L) excels in long-tail scenarios, while Vittle(F) performs better under perturbations. Since in many benchmarks either Vittle(F) or Vittle(L) brings very limited or even negative performance gain (e.g., Vittle(F) in Table 2 and Vittle(L) in Figure 5 left), it seems Vittle requires task-specific hyper-param choice (F or L) under distribution shift.
>
> We appreciate the reviewer’s insightful observation and agree that Vittle (F) and (L) emphasize different aspects of robustness. This is expected by design: the fixed prior (F) enforces stronger compression, yielding greater invariance and robustness under perturbations, while the learnable prior (L) introduces more flexibility, which benefits long-tail and knowledge-intensive tasks. Importantly, across 45 datasets and 30 shift scenarios, both variants can improve robustness relative to the baseline in most cases, and the magnitude of negative regressions is small. Thus, **rather than requiring task-specific tuning, the two variants represent different robustness–adaptability trade-offs**: (F) is preferable when robustness to perturbations is critical, while (L) is preferable when generalization to diverse long-tail queries is important. We will clarify this trade-off more explicitly in the paper and note that developing a unified adaptive prior that balances both regimes is an exciting direction for future work.
>
>
> > W2: The relationship between IB and model robustness in distribution shift is not clearly stated. Why "This property is particularly 129 desirable for robust instruction tuning"(Line 128-129)? Can IB enhance the robustness of other models in other tasks (e.g., CNNs in image classification) under distribution shift?
>
> The connection between IB and robustness is that IB explicitly encourages representations to discard nuisance variability while retaining task‑relevant information, which naturally mitigates sensitivity to distribution shifts. In the instruction‑tuning context, this is particularly desirable because multimodal inputs often contain spurious or modality‑specific noise (e.g., background clutter, token irregularities). **Our theoretical analysis** using the effective mutual information difference (EMID) **further formalizes this intuition by showing that stronger compression yields a tighter bound on the train–test performance gap**.
>
> We would also like to note that we already discussed IB’s effectiveness beyond MLLMs in Related Work (**L104–L117**), citing applications in CNNs and other architectures where IB improves robustness under corruptions and domain shifts. Other explicit examples of IB to enhance robustness under distribution shifts include DIB [Dubois et al. 2020] and IIB [Li et al. 2022]. Our contribution is to bring this principle to autoregressive multimodal instruction tuning, where robustness under distribution shift has been largely unexplored, and to provide both theoretical justification and large‑scale empirical validation.
>
>
> For this rebuttal, we further conducted an experiment on a ViT-Base ImageNet fine-tuning setup for image classification task with the same configuration adopted in Wortsman et al. 2022 (i.e. fine-tuning of a CLIP pre-trained ViT over 10 epochs on ImageNet).
>
> | Fine-tune Loss          | ImageNet (Clean) | ImageNet-C (Nine perturb. Avg) |
> |---------------|------------------|--------------------------------|
> | Cross-Entropy | 81.14            | 54.31                          |
> | w/ Bottleneck | **81.52**            | **56.19**                          |
>
> We observe that information bottleneck is still beneficial to improve robustness on this unimodal image classification setup with a pure vision model.
>
> > W3: Vittle introduces 20% extra training time and 1.5% extra parameters compared to baseline. Concerning that in many benchmarks Vittle brings very limited performance gain, is it possible that the difference simply comes from the increase of the training time and parameters? I think it is necessary for the authors to clarify this problem.
>
> We thank the reviewer for raising this important point. We agree that it's important to understand this more clearly.
>
> First, the added parameter count is only 1.5%, localized to the bottleneck module, which is negligible compared to the 7B–13B LLM backbone; prior work shows such a small increase has no measurable effect on capacity. Second, we controlled for training time by conducting longer‑epoch baselines: extending baseline training by the same 20% brings no comparable robustness improvements to Vittle's improvements. Finally, our method preserves in‑distribution accuracy, unlike other regularization approaches with similar or higher cost (e.g., weight decay, information‑maximization baselines; Table 3 & Fig. 6).
>
> | Epoch | Method     | POPE V Shift |
> |-------|------------|--------------|
> | 1     | Baseline   |      83.38        |
> | 1.2 (20% longer)   | Baseline   | 84.17             |
> | 1     | Vittle (F) |      **84.74**        |
>
> Thus, **the observed robustness improvements arise from the information‑theoretic bottleneck regularization introduced by Vittle, not from marginal increases in parameters or training time**. We will clarify this in the revision.
>
> ### Reference
> - Dubois et al. 2020, Learning Optimal Representations with the Decodable Information Bottleneck
> - Li et al. 2022, Invariant Information Bottleneck for Domain Generalization
> - Wortsman et al. 2022, Robust fine-tuning of zero-shot models

---

> > ### Author Response · Authors · 2025-08-06
> >
> > Dear Reviewer haEx,
> >
> > Thank you again for your productive review. We have prepared a response to address the points that you raised, and are wondering whether your concerns are addressed.
> > We appreciate your time commitment and consideration again!
> >
> > Best regards,
> >
> > The authors

---

> > ### Comment · Reviewer_haEx · 2025-08-07
> >
> > Thanks for the rebuttal. The experiments in W2 and W3 are convincing. I will keep my score.

---

### Official Review · Reviewer_eMVr · 2025-07-01

**Clarity:** 3
**Significance:** 3
**Originality:** 3
**Rating:** 5
**Confidence:** 4

**Summary:**

This paper proposes a new approach, called Visual Instruction Bottleneck Tuning, to improve the robustness of multimodal large language models under distribution shifts. Instead of scaling data or model, they apply the Information Bottleneck (IB) principle to encourage the model to learn minimal sufficient representations. They derive a novel variational lower bound for the IB objective tailored for MLLMs, and instantiate it as a simple bottleneck layer inserted into the LLM backbone. They test Vittle on 45 datasets, showing consistent gains on various perturbation and long-tail settings, while maintaining strong performance on standard benchmarks.

**Questions:**

1. Did you try applying Vittle on purely language tasks (without images)? Will the IB principle still help for textual instruction tuning?

2. How stable are your results across different random seeds? Given that sampling is used in bottleneck, is variance significant?

3. Could you share intuition why the fixed standard Gaussian prior Vittle(F) sometimes works better under perturbations than the learnable prior?

**Ethical Concerns:**

["NO or VERY MINOR ethics concerns only"]

**Final Justification:**

Most of my concerns are addressed. I will keep my previous score.

**Quality:**

3

**Strengths And Weaknesses:**

**Strengths**

1. The paper addresses a very practical and important problem, i.e., the robustness of MLLMs under distribution shifts. Although there exists several research works on this domain, it is still largely under-explored to me.

2. The idea to integrate IB principle into end-to-end instruction tuning for large-scale MLLMs is novel and well motivated.

3. I appreciate the theoretical analysis, especially how they connect their IB objective to EMID robustness metric.

4. Experiments are very extensive, covering various types of distribution shifts and several long-tail benchmarks. The improvements are quite consistent

5. The implementation is lightweight, only adds ~1.5% parameters, with nearly unchanged inference cost.

**Weakness**

1. The method requires tuning hyperparameters like α. It is unclear how sensitive final performance is to this.

2. The paper focuses mainly on visual QA tasks. It is interesting to see whether the same approach can benefit more realistic tasks like agentic tasks.

---

> ### Author Rebuttal · Authors · 2025-07-30
>
> The authors sincerely appreciate your comments and are happy to hear that you find a couple of strengths in our work! We provide a response to the pointed out weaknesses (W) and questions (Q) item by item.
>
> > W1: The method requires tuning hyperparameters like $\alpha$. It is unclear how sensitive final performance is to this.
>
> In **Table 10 of Appendix**, we presented the ablation study for the $\alpha$ parameter (along with $\beta$ and bottleneck insertion layer $l$ in Figure 12). As we can see in the table below, Vittle consistently outperforms the baseline, indicating that the method is generally robust to this choice. However, when $\alpha$ is increased beyond 0.5, we observed unstable training and exploding losses. This suggests that over‑relying on bottleneck‑generated representations can harm the stability of the instruction tuning phase, and thus $\alpha$ should be set conservatively.
>
> | Alpha           | POPE V Shift |  LB-COCO T Shift |
> |-----------------|--------------|-----------------|
> | baseline (0)    |  84.12        |  72.3            |
> | Vittle L (0.1)  |  84.20        |  73.1            |
> | Vittle L (0.25) |  84.47        |  73.1            |
> | Vittle L (0.5)  | 84.90        |  73.0             |
>
>
> > W2: The paper focuses mainly on visual QA tasks. It is interesting to see whether the same approach can benefit more realistic tasks like agentic tasks. / Q1: Did you try applying Vittle on purely language tasks (without images)? Will the IB principle still help for textual instruction tuning?
>
> We appreciate this thoughtful suggestion! Our manuscript focused on multimodal instruction tuning---an increasingly popular paradigm---where robustness to visual and textual perturbations is most critical. That said, we agree that extending Vittle to broader domains, such as agentic tasks and purely unimodal setups, is an exciting direction. In principle, the IB framework is modality‑agnostic, since it regularizes representations to discard nuisance factors while retaining task‑relevant information.
>
>
> To substantiate this, we are performing two additional unimodal experiments during the rebuttal:
>
>
> 1. **Language‑only setup**: We trained GPT‑2‑small [Brown et al. 2020] on FineWebEdu‑10B [Penedo et al. 2024] and evaluate the model on the clean and perturbed version of the HellaSwag validation split.
> 2. **Vision‑only setup**: We fine-tune a pre-trained ViT-Base on ImageNet by following the configs of Wortsman et al. 2022, and evaluate the model on the clean and perturbed version of the ImageNet validation split.
>
> | Fine-tune Loss          | ImageNet (Clean) | ImageNet-C (Nine perturb. Avg) |
> |---------------|------------------|--------------------------------|
> | Cross-Entropy | 81.14            | 54.31                          |
> | w/ Bottleneck | **81.52**            | **56.19**                          |
>
> Unfortunately, we have yet to obtain results on Language-only setup due to resource and time constraints, but we will keep you updated once the results are available. However, the promising result on the vision model fine-tuning setup implies that our proposed recipe of the information bottleneck-based training is a general method that can be applied to diverse setups of large-scale AI modeling not only for visual instruction tuning.
>
>
> > Q2: How stable are your results across different random seeds? Given that sampling is used in bottleneck, is variance significant?
>
> We thank the reviewer for this question. As suggested, we repeated key experiments with **three random seeds**, and report the aggregated performance for these runs below. Although Vittle shows some variation in final performance depending on the seed, the variance is not significant compared to the baseline’s own seed variance in terms of the coefficient of variation (CV), which is defined as STD/Mean to measure scale-normalized variance.
>
> | Method     | Mean ($\mu$)   | STD ($\sigma$)    | CV (${\sigma\over\mu}$)    |
> |------------|--------|--------|--------|
> | Baseline   | 83.44 | 0.18 | 0.22 |
> | Vittle (F) | 84.84 | 0.21 | 0.24 |
>
> > Q3: Could you share intuition why the fixed standard Gaussian prior Vittle(F) sometimes works better under perturbations than the learnable prior?
>
> Both fixed (F) and learnable (L) Gaussian priors of Vittle are **instance-independent priors** to enforce the bottleneck principle, but the **(L) prior is data-dependent** because it learns the distributional parameters on the entire training dataset (although it does not have an instance-level dependency), whereas the **(F) prior with standard Gaussian does not make any dependency on data**. Therefore, _the posterior distribution of input representations from Vittle (F) is more strongly compressed to decrease the KL divergence between posterior and prior than that of Vittle (L)._
>
> Intuitively, the more compressive posterior representations imply stronger invariance to perturbations, and based on our theoretical analysis, **the more compressive representation induces the tighter upper bound of the effective mutual information difference (EMID)**, a measure of robustness, if the train and test input distributions share the support.
>
>
>
> ### Reference
> - Brown et al. 2020, Language Models are Few-Shot Learners
> - Penedo et al. 2024, The FineWeb Datasets: Decanting the Web for the Finest Text Data at Scale
> - Wortsman et al. 2022, Robust fine-tuning of zero-shot models

---

> ### Author Response · Authors · 2025-08-05
>
> Dear Reviewer eMVr,
>
> We want to express our sincere appreciation again for your helpful feedback that contributed to enhancing the quality of this work, and we are happy to know that you found several strengths in our paper, including the importance of the problem statement, the novelty of our method with strong motivation, EMID-based theoretical analysis, and promising empirical results. Also, thanks for acknowledging our rebuttal. If you have any follow-up questions, feel free to let us know, and we will be happy to discuss further!
>
> Thank you again for taking the time to review our paper. We feel fortunate to have had you as our reviewer, who gave us such professional feedback.
>
> Best regards,
> The authors

---

> > ### Comment · Reviewer_eMVr · 2025-08-06
> > **Official Comment**
> >
> > Thanks for the rebuttal. I will keep my score.

---

### Official Review · Reviewer_wdcV · 2025-07-02

**Clarity:** 3
**Significance:** 2
**Originality:** 2
**Rating:** 4
**Confidence:** 3

**Summary:**

This paper introduces the Information Bottleneck (IB) theory into end-to-end instruction tuning for Multimodal Large Language Models (MLLMs), aiming to improve model robustness to out-of-distribution inputs without increasing data volume or model size. The authors derive a variational IB lower bound tailored for autoregressive MLLMs, design lightweight and pluggable bottleneck layers, and validate the method across 30 distribution shift scenarios, demonstrating consistent improvements in both open-/closed-ended QA and object hallucination detection tasks.

**Questions:**

Please refer to the weakness.

**Ethical Concerns:**

["NO or VERY MINOR ethics concerns only"]

**Final Justification:**

I appreciate the authors' efforts in addressing my concerns. I have carefully read the rebuttal. I will keep my score.

**Limitations:**

Yes

**Quality:**

3

**Strengths And Weaknesses:**

Strengths:

1. Simple and general implementation: A single pluggable MLP layer with two modes—fixed or learnable priors—allows reusability in existing training pipelines.

2. Comprehensive experiments: The study covers three task types, 45 datasets, and 27 perturbation levels, including ablations, prior comparisons, and fine-grained severe shift analyses. Results consistently outperform standard instruction tuning and several baselines (e.g., LoRA, weight decay, information maximization methods).

3. Clear writing and logical structure.

Weaknesses:

1. Mild performance regressions underexplored: Slight drops occur in some multi-disciplinary multiple-choice tasks (e.g., MMMU), but causes and trade-offs are not discussed.

2. Lack of safety and bias analysis: Information compression may risk filtering critical information (e.g., leading to misdiagnoses), yet the paper does not explore or assess such potential downsides.

3. Limited resource consumption reporting: Only training/inference time is reported, with no details on memory or GPU usage.

---

> ### Author Rebuttal · Authors · 2025-07-30
>
> Thanks for taking your time to provide this constructive feedback! We provide a response to the pointed out weaknesses (W) item by item.
>
> > W1: Mild performance regressions underexplored: Slight drops occur in some multi-disciplinary multiple-choice tasks (e.g., MMMU), but causes and trade-offs are not discussed.
>
> We thank the reviewer for this insightful observation. We believe the observed regressions can be explained by the nature of the bottleneck principle: while it enhances robustness by reducing sensitivity to superficial perturbations, it may also filter out fine-grained visual details that are not explicitly anchored by instruction tokens. Since MMMU includes many OCR-related, knowledge-intensive splits (e.g., Accounting, Finance), this trade-off can lead to slight drops in performance. Notably, the learnable-prior variant, Vittle (L), mitigates this issue by adapting its prior to retain more task-relevant details, **recovering performance closer to the non-bottleneck baseline** (35.3 vs 35.6, Table 2). We view this as evidence that the prior design offers a lever to balance robustness against general VQA benchmark performance, and we will make sure to add this discussion in our manuscript.
>
>
> > W2: Lack of safety and bias analysis: Information compression may risk filtering critical information (e.g., leading to misdiagnoses), yet the paper does not explore or assess such potential downsides.
>
> We agree that it is important to analyze the potential downsides of our method to ensure its reliable use in real-world applications. By reflecting Reviewer wdcV’s concern, we conducted three more experiments to evaluate the **output safety** with MM-SafetyBench [Liu et al. 2023], **fine-granular visual recognition capability** with OCRBench v2 [Fu et al. 2025], and output **textual diversity** [Li et al. 2015; Shaib et al. 2024].
>
> | Method     | MM-SafetyBench (&darr;) | OCRBench v2 Acc (&uarr;) | Distinct-1 (&uarr;) | Distinct-2 (&uarr;) | CR (&darr;) | Hom RL (&darr;) |
> |------------|---------------------|---------------|------------|------------|-------|-----------|
> | Baseline   | 0.6667              | **0.286**         | 0.2356     | 0.6117     | 3.234 | 0.155     |
> | Vittle (L) | **0.6026**              | 0.285          | **0.2413**     | **0.6279**     | **3.212** | 0.147     |
> | Vittle (F) | 0.6821              | 0.280         | 0.2382     | 0.6260      | 3.238 | **0.144**     |
>
>
> We observe that (1) under jailbreak prompting, Vittle (L) induces stronger refusal and achieves better safety than the baseline, whereas Vittle (F) slightly underperforms the baseline; (2) as hypothesized in the response to W1, both Vittle (L) and (F) marginally hurt fine-grained visual recognition capability compared to the baseline; but (3) both Vittle (L) and (F) show higher output textual diversity across four measures overall.
>
>
> > W3: Limited resource consumption reporting: Only training/inference time is reported, with no details on memory or GPU usage.
>
> We agree that a more complete resource analysis is valuable. In addition to the training/inference time reported in Table 5, we have now measured GPU memory usage. In the table below, we report the per-device maximum peak memory allocation during training and inference and the total GPU hours (single-gpu-basis) as below.
>
> | Method   | Peak Mem GB (train) | Peak Mem GB (test) | Total GPU hours (train) |
> |----------|---------------------|--------------------|-------------------------|
> | Baseline | 37.55               | 15.62              | 88                      |
> | Ours     | 38.98               | 15.84              | 106                     |
>
> Overall, the results confirm that Vittle improves robustness with minimal inference time computational and memory overhead, making it practical for deployment.
>
> ### Reference
> - Liu et al. 2023, MM-SafetyBench: A Benchmark for Safety Evaluation of Multimodal Large Language Models
> - Fu et al. 2025, OCRBench v2: An Improved Benchmark for Evaluating Large Multimodal Models on Visual Text Localization and Reasoning
> - Li et al. 2015, A Diversity-Promoting Objective Function for Neural Conversation Models
> - Shaib et al. 2024, Standardizing the Measurement of Text Diversity: A Tool and a Comparative Analysis of Scores

---

> ### Author Response · Authors · 2025-08-06
>
> Dear Reviewer wdcV,
>
> Thanks again for taking the time to review our paper and giving helpful feedback that is crucial to improving the quality of our paper. We are delighted to hear that you found several strengths in our work, including the simple and general implementation, comprehensive experiments, and clear and logical writing structure. We also appreciate that you have acknowledged our rebuttal. If you have any additional opinions or questions, do not hesitate to let us know!
>
> Best regards, The authors

---

### Official Review · Reviewer_RnJj · 2025-07-09

**Clarity:** 3
**Significance:** 3
**Originality:** 3
**Rating:** 3
**Confidence:** 3

**Summary:**

The paper introduces Visual Instruction Bottleneck Tuning (Vittle), which inserts an information-bottleneck layer inside the LLM backbone to regularise representations. Robustness is evaluated on 30 distribution-shift scenarios created from POPE (object-hallucination) and LB-COCO (open-ended QA) perturbation suites.

**Questions:**

Questions for Authors

1. Does Vittle still help when applied to recent models?

2. How sensitive is performance to the IB weight β and to the insertion layer l?

3. Why limit the bottleneck to instruction-tuning? Would pre-training with IB amplify robustness?

4. Could the authors elaborate on when Vittle (L) vs (F) is preferable?

5. Have you considered more realistic perturbations (partial occlusion, codec compression, ASR/OCR noise)?

6. What is the inference-time latency and memory overhead beyond the ~1.5 % parameter increase and 20 % training-time rise?

7. Does the bottleneck reduce answer diversity or expressiveness while lowering hallucination?

8. Please clarify concrete advancements over the previous ICML-Workshop submission (data size, theory, experiments, scope).

These points should help strengthen your review and guide the discussion phase.

**Ethical Concerns:**

["NO or VERY MINOR ethics concerns only"]

**Final Justification:**

Dear authors,

During the rebuttal period, they addressed many of the points I raised to some extent, but I believe it is difficult to draw a conclusion based solely on the experiments conducted within the short rebuttal window. For the new vision–language models, most of the experiments presented in the paper should be reproduced, and the current perturbation should also be applied to all vision-related experiments in the paper. Therefore, I will maintain my current score.

**Limitations:**

yes

**Quality:**

2

**Strengths And Weaknesses:**

Strength:

1. Solid theoretical grounding. The authors derive a variational IB lower bound and link it to EMID, a robustness metric.

2. First end-to-end IB for MLLMs. Unlike prior projector-only work, Vittle places a learnable bottleneck within the LLM and trains it jointly.

3. Clarity of exposition. The paper is well-structured with intuitive diagrams and ablation details.

Weaknesses

1. Marginal gains. Relative improvements on perturbed sets are mostly single-digit pp; absolute scores remain close to the baseline.

2. Perturbation realism. Visual corruptions focus on generic CV transforms (brightness, blur, noise); they might not match real LMM deployment errors (occlusion, compression, camera shake).

3. Older model backbone. Experiments centre on LLaVA-v1.5-7B/13B and Prism-7B; results on newer models are missing.

4. Unclear novelty over ICML workshop version. The manuscript does not explicitly differentiate itself from the prior workshop paper.

---

> ### Author Rebuttal · Authors · 2025-07-30
>
> Thanks for taking the time to point out some important points that are worth discussing! Below, we provide responses to individual weaknesses (W) and questions (Q) you pointed out.
>
> > W1: Clarification on performance gains.
>
> We acknowledge the reviewer’s point, and would like to emphasize that **achieving improvements across 30 different types of distribution shifts and multiple models is non-trivial**, especially given the strong baseline performance. Vittle delivers significant improvements in many cases, such as LB-COCO text shifts – Word Insert, Char Typo, and Arabic; most of the LB-COCO joint shift cases. We would like to note that these improvements, i.e., **3.5% ~ 9% improvement over the baseline, are not quite trivial**, and the improvements are also consistent across multiple experimental settings, which demonstrates the effectiveness of our method.
>
> These robustness gains are particularly valuable as they come with only 1.5% parameter overhead and almost no inference-time cost increase. Importantly, Vittle also preserves performance on the original in-distribution datasets, ensuring that robustness does not come at the expense of standard task performance. We therefore believe the improvements represent meaningful enhancements for MLLMs.
>
> > W2, Q5: Considering more realistic shifts.
>
> We thank the reviewer for raising this important point. Our perturbation set was designed to **cover a wide range of standardized, controllable shifts commonly adopted in robustness evaluation** for CV and (M)LLMs (e.g., [Dan et al. 2019; Qiu et al. 2024; Oh et al., 2025]), allowing systematic comparison across multiple models and settings. While these generic corruptions (brightness, blur, noise) do not exhaustively capture all real-world deployment errors, they provide a principled testbed for stress-testing robustness in a reproducible manner. Importantly, our study does not rely solely on synthetic perturbations: we also evaluate on long-tail, naturally collected benchmarks such as LB-Wild, LB-Wilder, and WV-Bench, which **reflect realistic distribution shifts** in user queries (see **Section 4.2**). Vittle shows consistent gains on these datasets as well, suggesting that the improvements are not limited to artificial settings.
>
> We agree that occlusion, compression artifacts, and camera shake are important additional sources of shift. However, a lack of benchmark datasets on such realistic perturbations is a major obstacle on this path. Establishing a more realistic distribution shift benchmark for MLLMs is therefore an important direction for future work. Nonetheless, the current results already demonstrate that Vittle improves robustness across both controlled perturbations and naturally occurring distributions.
>
>
> > W3, Q1: Applicability to recent models.
>
> Our primary goal was not to chase state-of-the-art performance, but to demonstrate the effectiveness of the proposed training paradigm in a **standard, widely adopted setup that ensures fair comparison and easy reproducibility**. For this reason, we focused on the well-established LLaVA‑v1.5 training pipeline, which is the common benchmark in prior work [Wang et al., 2025; Zhou et al., 2025], and also reported results with one recent model Prism [Karamcheti et al., 2024] in the Appendix.
>
> That said, we agree it is important to validate generality on newer models. To this end, we additionally evaluated Vittle on `LLaVA‑Mini` [Zhang et al., 2025], a recently proposed MLLM with an efficiency‑oriented compression module. Using the same configuration as LLaVA‑Mini (Vicuna‑7B backbone, Table 1 in Zhang et al. 2025), we observed that Vittle improved robustness on POPE visual shifts while preserving in‑distribution performance.
>
> | Method     | POPE  | POPE V Shift |
> |--|--|--|
> | Baseline   | 79.37 | 77.39        |
> | Vittle (F) | **81.07** | **78.32**        |
>
> More broadly, given the rapid pace of MLLM development, it is inevitable that even newer models will emerge after the completion of this work. We emphasize that our contribution is a training framework that is designed to be **model‑agnostic** and easily applicable to alternative backbones.
>
> > W4, Q8: Novelty over ICML workshop version.
>
> We would like to clarify that the ICML workshop is a **non-archival workshop without proceeding, and thus the NeurIPS submission should be evaluated independently on its own merits**. Workshop is mostly intended to share early ideas with the community. Based on the submission policy noted in Call for Paper NeurIPS 2025, papers presented at workshops (without proceeding) are permitted, whether they are substantially similar to the submitted version or not. Therefore, the policy does not warrant the need to claim the novelty over workshop version. We thank you for checking on this.
>
> > Q2: How sensitive is performance to the IB weight β and to the insertion layer l?
>
> We provide a sensitivity analysis of both the IB weight $\beta$ and the insertion layer $l$ in **Fig. 12 of Appendix**. We find that the choice of insertion layer plays a vital role in ensuring stable performance across diverse tasks. This aligns with findings from Skean et al. (2025), who showed that the selection of probing layers in LLMs strongly influences downstream outcomes; we observe a similar trend in the visual instruction tuning phase of Vittle. The IB weight $\beta$ also affects performance, as values that are too small or too large can destabilize training, but overall, $\beta$ is less sensitive than the choice of insertion layer.
>
> > Q3: Would pre-training with IB amplify robustness?
>
> Thanks for the suggestion! Our chief goal of this work is to improve the robustness of MLLM to distribution shifts. However, it is hard to define “distribution shifts” for the pre-trained model because most of the recent MLLMs have undergone a hyper-scale pre-training with tons of data points.
>
> In contrast, instruction tuning is conducted on a relatively small scale with a limited number of data points, which enables us to define the distribution shift between train and test well and helps us to derive a clear problem statement. Therefore, we confine our interest to the instruction tuning setup in our manuscript.
>
> Investigating the effectiveness of IB in a pre-training setup is also an important direction. Although we are currently conducting an experiment on GPT-2 pre-training, we have yet to obtain results due to resource and time constraints, but we will keep you updated once the results are available.
>
> > Q4: Could the authors elaborate on when Vittle (L) vs (F) is preferable?
>
> As discussed in **Section 4.2**, Vittle (F) tends to perform better under perturbations due to its stronger isotropic prior constraint, making it more robust to distribution shifts. In contrast, Vittle (L) leverages a learnable prior, which provides greater adaptability and yields stronger performance on long-tail queries and knowledge-intensive tasks without perturbations. In practice, we recommend using Vittle (F) when robustness to severe test-time perturbations is the priority, and Vittle (L) when adaptability to diverse, long-tail samples and preservation of general knowledge are more critical.
>
>
> > Q6: What is the inference-time latency and memory overhead?
>
> - In **Table 5** of our manuscript, we already report the inference-time latency comparison of our method with the baseline, and show that **Vittle achieves almost identical inference time with the baseline**.
> - In terms of memory overhead, we report train and inference-time peak memory below. As we can see, Vittle has almost comparable peak memory usage.
>
> | Method   | Peak Mem GB (train) | Peak Mem GB (test) |
> |----|-----|------|
> | Baseline | 37.55               | 15.62              |
> | Ours     | 38.98               | 15.84              |
>
> > Q7: Does the bottleneck reduce answer diversity or expressiveness?
>
> To investigate the output text diversity with and without bottleneck, we evaluate a common textual diversity metric Distinct n-gram [Li et al. 2015], as well as compression ratio (CR) and Rouge-L-based homogeneity score (Hom RL) [SShaib et al. 2024] over outputs from each model on LB-COCO clean dataset.
>
> | Method     | Distinct-1 (&uarr;) | Distinct-2 (&uarr;) | CR (&darr;) | Hom RL (&darr;) |
> |---|-----|-----|----|----|
> | Baseline   | 0.2356     | 0.6117     | 3.234 | 0.155     |
> | Vittle (L) | **0.2413**     | **0.6279**     | **3.212** | 0.147     |
> | Vittle (F) | 0.2382     | 0.6260     | 3.238 | **0.144**     |
>
> The result implies that Vittle does not hurt output diversity compared to the baseline and even improves it, achieving a favorable balance between invariance and sensitivity in the inner representation space to produce better responses per input.
>
>
> ### Reference
> - Dan et al. 2019, Benchmarking Neural Network Robustness to Common Corruptions and Perturbations
> - Qiu et al. 2024, Benchmarking Robustness of Multimodal Image-Text Models under Distribution Shift
> - Oh et al. 2025, Understanding Multimodal LLMs Under Distribution Shifts: An Information-Theoretic Approach
> - Wang et al. 2025, Reconstructive Visual Instruction Tuning
> - Zhou et al. 2025, Learning to Instruct for Visual Instruction Tuning
> - Karamcheti et al. 2024, Prismatic VLMs: Investigating the Design Space of Visually-Conditioned Language Models
> - Zhang et al. 2025, LLaVA-Mini: Efficient Image and Video Large Multimodal Models with One Vision Token
> - Skean et al. 2025, Layer by Layer: Uncovering Hidden Representations in Language Models
> - Brown et al. 2020, Language Models are Few-Shot Learners
> - Penedo et al. 2024, The FineWeb Datasets: Decanting the Web for the Finest Text Data at Scale
> - Zellers et al. 2019, HellaSwag: Can a Machine Really Finish Your Sentence?
> - Li et al. 2015, A Diversity-Promoting Objective Function for Neural Conversation Models
> - Shaib et al. 2024, Standardizing the Measurement of Text Diversity: A Tool and a Comparative Analysis of Scores

---

> > ### Author Response · Authors · 2025-08-06
> >
> > Dear reviewer RnJj,
> >
> > Thank you once again for taking the time to review our paper and providing valuable comments.
> >
> > Following your feedback, we have worked diligently to address your concerns and clarified the questions you've raised. In this process, we have obtained  additional supporting evidence by addressing questions from you and other reviewers, such as applicability to other backbone (RnJj), stability analysis (eMVr), IB for vision model fine-tuning (haEx), complementary effect with data augmentation (BHRh), and dual-stream VLM applicability (BHRh), which demonstrates the generality of our method.
> >
> >
> > As the discussion deadline approaches, please don't hesitate to let us know if you have any additional questions. We'd be happy to respond and are committed to revising the paper to reflect your suggestions and improve its quality.
> >
> > Thank you for your time and consideration.
> >
> > Best regards,
> > The authors

---

> > > ### Comment · Reviewer_RnJj · 2025-08-07
> > >
> > > Thank you for the detailed responses. Unfortunately, the core concerns I raised remain only partially addressed:
> > >
> > > W2 – Perturbation realism
> > >
> > > Even acknowledging the absence of public occlusion / compression benchmarks, I am still skeptical that the chosen synthetic corruptions capture realistic deployment errors. Concrete plans or preliminary experiments toward more realistic shifts would have helped.
> > >
> > > W3 – Validation on recent backbones
> > >
> > > Evaluating on LLaVA-Mini does not alleviate the concern, as it is itself based on an older Vicuna-7B backbone. Without evidence on truly up-to-date models (e.g., recent Gemini/Llama-3-V or similar), the generality of Vittle remains uncertain.
> > >
> > > W4 – Differences from the ICML workshop version
> > >
> > > You did not answer my question. I asked you for the difference between the ICML version and this version. The rebuttal explained the non-archival status of the workshop paper, but did not enumerate what is actually new in this submission (additional data, theory, or experiments).
> > >
> > > Given these issues, I would like to keep my original scores.

---

> ### Author Response · Authors · 2025-08-09
> **Follow-up Response to Reviewer RnJj**
>
> Dear Reviewer RnJj,
>
> We thank you participation in the discussion and for clarifying your remaining concerns. Below, we respectfully provide responses to your comments `W2`, `W3`, and `W4`.
>
> > `W2` - Perturbation realism and `W3` - Validation on recent backbones
>
> We appreciate your point on the realistic perturbation benchmarks [`W2`]. By reflecting on your suggestions, **we conducted additional experiments to simulate 1) camera shake and 2) OCR noise you mentioned in your initial review.** For the camera shake, we leverage `wandlibrary.MagickMotionBlurImage` of the `wand` Python package to simulate fast camera movement (shaking). For the OCR noise, we apply a typographic attack similar to the setting of [OpenAI 2021] where a single word on top of a white rectangular patch is randomly attached to an image. Here we pick the most effective word, "groenendael", for attack by following [Wang et al. 2025]. We leverage the POPE dataset as our target and then generate perturbed/attacked versions of POPE.
>
>
> Here, we experimented with **Llama-3-8B-Instruct (which is the language model backbone of Llama-3-V that you mentioned) to address your concern on up-to-date backbone applicability [`W3`]** by following the training configuration of Rasheed et al. 2024, one of the popular community-driven open-source implementations, LLaVA++ (>800 GitHub stars). We set all the hyperparameters, such as learning rate and weight decay same values as their recommendation running script, except the batch size due to memory constraints. Note that Gemini (another model you have mentioned) experiment is not applicable since it's a closed-source model.
>
> For Vittle's training configuration, we set $\alpha=0.25$, $\beta=0.5$, and the bottleneck insertion layer $l=26$, and summarize the results as below.
>
> | Language Model Backbone | Method   | POPE V Shift Avg. | Camera Shake | OCR Attack (universal text patch) |
> |------|----------|-----------|---------|------------|
> | Llama-3-8B-Instruct     | Baseline |80.54|       80.73       |85.33|
> | Llama-3-8B-Instruct     | Ours     |**84.08**|**81.53**|**85.93**|
>
> * As shown, **Vittle consistently outperforms the baseline framework under these realistic perturbations, as well as the standard corruptions** used in our manuscript.
> * Moreover, **Vittle demonstrates its generality on this newer backbone (Llama-3) as well**.
>
>
> > `W4` - Differences from the ICML workshop version
>
> We apologize for not addressing this concern clearly in our initial response. To directly answer your question: _there are no substantive differences between the ICML workshop version and this NeurIPS submission._ We deliberately refrained from explicitly citing the workshop version or including a statement such as “there are no differences from the workshop version [CITATION]” in the manuscript, **since doing so would compromise the double-blind policy by revealing author identities**.
>
>
> Importantly, this practice—submitting the same work to a non-archival workshop and a top-tier conference—is **common and accepted within the ML community**, especially when used to gather early feedback ahead of formal peer review. Because the ICML workshop is non-archival, this dual submission is permitted under the NeurIPS 2025 Call for Papers.
>
> We hope this clarifies the situation, and we respectfully suggest that the lack of differences should not be viewed as a weakness, given that this approach is both standard and policy-compliant (one should regard this just as the same as a paper previously uploaded on ArXiv and then submitted to a conference).
>
> ---
>
> ### Final remark
>
> We would like to thank you again for your extensive feedback and the time you have invested. We hope that the additional experiments and clarifications provided here adequately address your remaining concerns.
>
> ### Reference
> - OpenAI 2021, Multimodal neurons in artificial neural networks
> - Wang et al. 2025, Typographic Attacks in a Multi-Image Setting
> - Rasheed et al. 2024, LLaVA++: Extending Visual Capabilities with LLaMA-3 and Phi-3

---

### Note · Authors · 2025-08-12

We sincerely thank all reviewers and chairs for their time and constructive engagement. Across all reviews, there was consistent recognition of the paper’s **solid theoretical foundation**, its **novelty** as the first end-to-end IB framework for MLLMs, the **clarity** of its exposition, the **breadth and rigor of its experimental validation**, and its direct relevance to a **practical and important problem**.

For convenience, we summarize below the reviewer stance, followed by the additional experiments and evidence we provided during the rebuttal.

---
### Post-rebuttal reviewer stance:

- Reviewer eMVr: Confirmed all concerns resolved and **kept positive rating for acceptance** (5).
- Reviewer haEx: Expressed that additional experiments are convincing, **maintaining a positive stance** (4).
- Reviewer BHRh: Confirmed all concerns resolved and **recommended acceptance** (5).
- Reviewer wdcV: Acknowledged the rebuttal, **maintaining a positive stance** (4).
- Reviewer RnJj: Questioned perturbation realism, backbone, and novelty over the workshop version. **We directly addressed all three remaining concerns with additional experiments and clarifications on Aug 8**.

---
### Summary of additional experiments and clarifications
During the rebuttal, we have:

1. Added experiments on (1) a recent backbone, (2) two-stream VLM, and (3) three data augmentation strategies, confirming general applicability across architectures and conditions.
2. Evaluated under two more realistic perturbations (camera shake, OCR noise).
3. Conducted additional analysis on safety and output diversity.
4. Extended validation beyond VQA to the image classification task.
5. Demonstrated stability results across different random seeds.
6. Added ablation on longer training time of baseline.
7. Clarified the relationship to the ICML workshop version, confirming compliance with NeurIPS’s dual-submission policy.

We believe these results address all substantive concerns and further strengthen the manuscript’s rigor and real-world relevance. We will incorporate these improvements and clarifications into the final version to ensure the work serves as a clear and reproducible reference for future research.

Thanks once again for your hard work. We hope this remark helps your discussion.

---

### Decision · Program_Chairs · 2025-09-17

**Decision:**

Accept (poster)

**Comment:**

This paper addresses the robustness of multimodal large language models under distribution shifts, by introducing visual instruction bottleneck tuning. The proposed method is grounded in the information bottleneck framework. The paper derives a variational lower bound, connects it to robustness metrics, and implements a practical algorithm. Experiments on several datasets across diverse shift scenarios show consistent improvements

This paper received the reviewing comments from five reviewers. Reviewers raised concerns in (i) unclear technical details and advancement over previous work; (ii) more fine-grained experimental result analysis (such as answer diversity, expressiveness, safety, and bias analysis); (iii) the balance between computational efficiency and performance gains; (iv) the difference between the current form and previous workshop version. After the rebuttal, four reviewers are positive and reach a consensus. Reviewer RnJj is concerned about the concern (iv). As this is common nowadays for submitting the same work to a non-archival workshop and a top-tier conference within the machine learning community, and does not violate the confidence policy, this is not considered to be the reason for the paper to be rejected.

The AC has checked the paper and the reviewers’ comments, and considers this to be solid work. Therefore, an acceptance recommendation is made. The authors are encouraged to incorporate the reviewers’ constructive feedback in the final version to further strengthen the clarity and impact of this work.